# PROVABLE MEMORIZATION VIA DEEP NEURAL NETWORKS USING SUB-LINEAR PARAMETERS

## ABSTRACT

It is known that $O(N)$ parameters are sufficient for neural networks to memorize arbitrary $N$ input-label pairs. By exploiting depth, we show that $O(N^{2/3})$ parameters suffice to memorize $N$ pairs, under a mild condition on the separation of input points. In particular, deeper networks (even with width 3) are shown to memorize more pairs than shallow networks, which also agrees with the recent line of works on the benefits of depth for function approximation. We also provide empirical results that support our theoretical findings.

## 1 INTRODUCTION

The modern trend of over-parameterizing neural networks has shifted the focus of deep learning theory from analyzing their expressive power toward understanding the generalization capabilities of neural networks. While the celebrated universal approximation theorems state that over-parameterization enables us to approximate the target function with a smaller error (Cybenko, 1989; Pinkus, 1999), the theoretical gain is too small to satisfactorily explain the observed benefits of over-parameterizing already-big networks. Instead of "how well can models fit," the question of "why models do not overfit" has become the central issue (Zhang et al., 2017).

Ironically, a recent breakthrough on the phenomenon known as the *double descent* (Belkin et al., 2019; Nakkiran et al., 2020) suggests that answering the question of "how well can models fit" is in fact an essential element in fully characterizing their generalization capabilities. In particular, the double descent phenomenon characterizes two different phases according to the capability/incapability of the network size for memorizing training samples. If the network size is insufficient for memorization, the traditional bias-variance trade-off occurs. However, after the network reaches the capacity that memorizes the dataset, i.e., "interpolation threshold," larger networks exhibit better generalization. Under this new paradigm, identifying the *minimum size of networks* for memorizing *finite* input-label pairs becomes a key issue, rather than function approximation that considers *infinite* inputs.

The memory capacity of neural networks is relatively old literature, where researchers have studied the minimum number of parameters for memorizing arbitrary $N$ input-label pairs. Existing results showed that $O(N)$ parameters are sufficient for various activation functions (Baum, 1988; Huang and Babri, 1998; Huang, 2003; Yun et al., 2019; Vershynin, 2020). On the other hand, Sontag (1997) established the negative result that for any network using analytic definable activation functions with $o(N)$ parameters, there exists a set of $N$ input-label pairs that the network cannot memorize. The sub-linear number of parameters also appear in a related topic, namely the VC-dimension of neural networks. It has been proved that there exists a set of $N$ inputs such that a neural network with $o(N)$ parameters can "shatter," i.e., memorize arbitrary labels (Maass, 1997; Bartlett et al., 2019). Comparing the two results on $o(N)$ parameters, Sontag (1997) showed that *not all* sets of $N$ inputs can be memorized for arbitrary labels, whereas Bartlett et al. (2019) showed that *at least one* set of $N$ inputs can be shattered. This suggests that there may be a reasonably large family of $N$ input-label pairs that can be memorized with $o(N)$ parameters, which is our main interest.

### 1.1 SUMMARY OF RESULTS

In this paper, we identify a mild condition satisfied by many practical datasets, and show that $o(N)$ parameters suffice for memorizing such datasets. In order to bypass the negative result by Sontag (1997), we introduce a condition to the set of inputs, called the $\Delta$-separateness.

**Definition 1.** *For a set $\mathcal{X} \subset \mathbb{R}^{d_x}$, we say $\mathcal{X}$ is $\Delta$-separated if*

$$\sup_{x,x' \in \mathcal{X}:x \neq x'} \|x - x'\|_2 < \Delta \times \inf_{x,x' \in \mathcal{X}:x \neq x'} \|x - x'\|_2.$$

This condition requires that the ratio of the maximum distance to the minimum distance between distinct points is bounded by $\Delta$. Note that the condition is milder when $\Delta$ is bigger. By Definition 1, any given finite set of (distinct) inputs is $\Delta$-separated for some $\Delta$, so one might ask why $\Delta$-separateness is different from having distinct inputs in a dataset. The key difference is that even if the number of data points $N$ grows, the ratio of the maximum to the minimum should remain bounded by $\Delta$. Given the discrete nature of computers, there are many practical datasets that satisfy $\Delta$-separateness, as we will see shortly. Also, this condition is more general than the minimum distance assumptions ($\forall i, \|x_i\|_2 = 1, \forall i \neq j, \|x_i - x_j\|_2 \geq \rho > 0$) that are employed in existing theoretical results (Hardt and Ma, 2017; Vershynin, 2020). To see this, note that the minimum distance assumption implies $2/\rho$-separateness. In our theorem statements, we will use the phrase "$\Delta$-separated set of input-label pairs" for denoting the set of inputs is $\Delta$-separated.

In our main theorem sketched below, we prove the sufficiency of $o(N)$ parameters for memorizing any $\Delta$-separated set of $N$ pairs (i.e., any $\Delta$-separated set of $N$ inputs with arbitrary labels) even for large $\Delta$. More concretely, our result is of the following form:

**Theorem 1** (Informal). *For any $w \in (\frac{2}{3}, 1]$, there exists a $O(N^{2-2w}/\log N + \log \Delta)$-layer, $O(N^w + \log \Delta)$-parameter fully-connected network with sigmoidal or* RELU *activation that can memorize any $\Delta$-separated set of $N$ input-label pairs.*

We note that $\log$ has base 2. Theorem 1 states that if the number of layers increases with the number of pairs $N$, then any $\Delta$-separated set of $N$ pairs can be memorized by a network with $o(N)$ parameters. Here, we can check from Definition 1 that the $\log \Delta$ term does not usually dominate the depth or the number of parameters, especially for modern deep architectures and practical datasets. For example, it is easy to check that any dataset consisting of 3-channel images (values from $\{0, 1, \dots, 255\}$) of size $a \times b$ satisfies $\log \Delta < (9 + \frac{1}{2} \log(ab))$ (e.g., $\log \Delta < 17$ for the ImageNet dataset), which is often much smaller than the depth of modern deep architectures.

For practical datasets, we can show that networks with parameters fewer than the number of pairs can successfully memorize the dataset. For example, in order to perfectly classify one million images in ImageNet dataset[1] with 1000 classes, our result shows that 0.7 million parameters are sufficient. The improvement is more significant for large datasets. To memorize 15.8 million bounding boxes in Open Images V6[2] with 600 classes, our result shows that only 4.5 million parameters suffice.

Theorem 1 improves the sufficient number of parameters for memorizing a large class of $N$ pairs (i.e., $2^{O(N^w)}$-separated) from $O(N)$ down to $O(N^w)$ for any $w \in (\frac{2}{3}, 1)$, for deep networks. Then, it is natural to ask whether the depth increasing with $N$ is *necessary* for memorization with a sub-linear number of parameters. The following existing result on the VC-dimension implies that this is indeed necessary for memorization with $o(N/\log N)$ parameters, at least for RELU networks.

**Theorem** [Bartlett et al. (2019)]. (Informal) *For $L$-layer* RELU *networks, $\Omega(N/(L \log N))$ parameters are necessary for memorizing at least a single set of $N$ inputs with arbitrary labels.*

The above theorem implies that for RELU networks of constant depth, $\Omega(N/\log N)$ parameters are necessary for memorizing at least one set of $N$ inputs with arbitrary labels. In contrast, by increasing depth with $N$, Theorem 1 shows that there is a large class of datasets that can be memorized with $o(N/\log N)$ parameters. Combining these two results, one can conclude that increasing depth is *necessary and sufficient* for memorizing a large class of $N$ pairs with $o(N/\log N)$ parameters.

Given that the depth is critical for the memorization power, is the width also critical? We prove that it is not the case, via the following theorem.

**Theorem 2** (Informal). *For a fully-connected network of width 3 with a sigmoidal or* RELU *activation function, $O(N^{2/3} + \log \Delta)$ parameters (i.e., layers) suffice for memorizing any $\Delta$-separated set of $N$ input-label pairs.*

Theorem 2 states that under $2^{O(N^{2/3})}$-separateness of inputs, the network width does not necessarily have to increase with $N$ for memorization with sub-linear parameters. Furthermore, it shows that

---

[1]http://www.image-net.org/
[2]https://storage.googleapis.com/openimages/web/index.html

even a surprisingly narrow network of width 3 has a superior memorization power than a fixed-depth network, requiring only $O(N^{2/3})$ parameters.

Theorems 1 and 2 show the *existence* of network architectures that memorize $N$ points with $o(N)$ parameters, under the condition of $\Delta$-separateness. This means that these theorems do not answer the question of how many such data points can a *given* network memorize. We provide generic criteria for identifying the maximum number of points given general networks (Theorem 3). In a nutshell, our criteria indicate that to memorize more pairs under the same budget for the number of parameters, a network must have a deep and narrow architecture at the final layers of the network. In contrast to the prior results that the number of arbitrary pairs that can be memorized is at most proportional to the number of parameters (Yamasaki, 1993; Yun et al., 2019; Vershynin, 2020), our criteria successfully incorporate with the characteristics of datasets, the number of parameters, and the architecture, which enable us to memorize $\Delta$-separated datasets with number of pairs super-linear in the number of parameters.

Finally, we provide empirical results corroborating our theoretical findings that deep networks often memorize better than their shallow counterparts with a similar number of parameters. Here, we emphasize that better memorization power does not necessarily imply better generalization. We indeed observe that shallow and wide networks often generalize better than deep and narrow networks, given the same (or similar) training accuracy.

**Organization.** We first introduce related works in Section 2. In Section 3, we introduce necessary notation and the problem setup. We formally state our main results and discuss them in Section 4. In Section 6, we provide empirical observations on the effect of depth and width in neural networks. Finally, we conclude the paper in Section 7.

## 2 RELATED WORKS

### 2.1 NUMBER OF PARAMETERS FOR MEMORIZATION

**Sufficient number of parameters for memorization.** Identifying the sufficient number of parameters for memorizing arbitrary $N$ pairs has a long history. Earlier works mostly focused on bounding the number of hidden neurons of shallow networks for memorization. Baum (1988) proved that for 2-layer STEP[3] networks, $O(N)$ hidden neurons (i.e., $O(N)$ parameters) are sufficient for memorizing arbitrary $N$ pairs when inputs are in general position. Huang and Babri (1998) showed that the same bound holds for any bounded and nonlinear activation function $\sigma$ satisfying that either $\lim_{x \to -\infty} \sigma(x)$ or $\lim_{x \to \infty} \sigma(x)$ exists, without any condition on inputs. The $O(N)$ bounds on the number of hidden neurons was improved to $O(\sqrt{N})$ by exploiting an additional hidden layer by Huang (2003); nevertheless, this construction still requires $O(N)$ parameters.

With the advent of deep learning, the study has been extended to modern activation functions and deeper architectures. Zhang et al. (2017) proved that $O(N)$ hidden neurons are sufficient for 2-layer RELU networks to memorize arbitrary $N$ pairs. Yun et al. (2019) showed that for deep RELU (or hard $\tanh$) networks having at least 3 layers, $O(N)$ parameters are sufficient. Vershynin (2020) proved a similar result for STEP (or RELU) networks with an additional logarithmic factor, i.e., $\widetilde{O}(N)$ parameters are sufficient, for memorizing arbitrary $\{x_i : \|x_i\|_2 = 1\}_{i=1}^N$ satisfying $\|x_i - x_j\|_2^2 = \Omega(\frac{\log \log d_{\max}}{\log d_{\min}})$ and $N, d_{\max} = e^{O(d_{\min}^{1/5})}$ where $d_{\max}$ and $d_{\min}$ denote the maximum and the minimum hidden dimensions, respectively.

In addition, the memorization power of modern network architectures has also been studied. Hardt and Ma (2017) showed that RELU networks consisting of residual blocks with $O(N)$ hidden neurons can memorize arbitrary $\{x_i : \|x_i\|_2 = 1\}_{i=1}^N$ satisfying $\|x_i - x_j\|_2 \geq \rho$ for some absolute constant $\rho > 0$. Nguyen and Hein (2018) studied a broader class of layers and proved that $O(N)$ hidden neurons suffice for convolutional neural networks consisting of fully-connected, convolutional, and max-pooling layers for memorizing arbitrary $N$ pairs having different patches.

**Necessary number of parameters for memorization.** On the other hand, the necessary number of parameters for memorization has also been studied. Sontag (1997) showed that for any neural network using analytic definable activation functions, $\Omega(N)$ parameters are necessary for memorizing arbitrary

---

[3]STEP denotes the binary threshold activation function: $x \mapsto \mathbf{1}[x \geq 0]$.

$N$ pairs. Namely, given any network using analytic definable activation with $o(N)$ parameters, there exists a set of $N$ pairs that the network cannot memorize.

The Vapnik-Chervonenkis (VC) dimension is also closely related to the memorization power of neural networks. While the memorization power studies the number of parameters for memorizing *arbitrary* $N$ pairs, the VC-dimension studies the number of parameters for memorizing *at least a single set* of $N$ inputs with arbitrary labels. Hence, it naturally provides the lower bound on the necessary number of parameters for memorizing arbitrary $N$ pairs. The VC-dimension of neural networks has been studied for various types of activation functions. For memorizing at least a single set of $N$ inputs with arbitrary labels, it is known that $\Theta(N/\log N)$ parameters are necessary (Baum and Haussler, 1989) and sufficient (Maass, 1997) for STEP networks. Similarly, Karpinski and Macintyre (1997) proved that $\Omega(\sqrt{N}/U)$ parameters are necessary for sigmoid networks of $U$ neurons. Recently, Bartlett et al. (2019) showed that $\Theta(N/(\bar{L}\log N))$ parameters are necessary and sufficient for $L$-layer networks using any piecewise linear activation function where $\bar{L} := \frac{1}{W_L}\sum_{\ell=1}^{L} W_\ell$ and $W_\ell$ denotes the number of parameters up to the $\ell$-th layer.

## 2.2 BENEFITS OF DEPTH IN NEURAL NETWORKS

To understand deep learning, researchers have investigated the advantages of deep neural networks compared to shallow neural networks with a similar number of parameters. Initial results discovered examples of deep neural networks that cannot be approximated by shallow neural networks without using exponentially many parameters (Telgarsky, 2016; Eldan and Shamir, 2016; Arora et al., 2018). Recently, it is discovered that deep neural networks require fewer parameters than shallow neural networks to represent or approximate a class of periodic functions (Chatziafratis et al., 2020a;b). For approximating continuous functions, Yarotsky (2018) proved that the number of required parameters for RELU networks of constantly bounded depth are square to that for deep RELU networks.

## 3 NOTATION AND PROBLEM SETUP

In this section, we introduce notation and the problem setup. We use $\log$ to denote the logarithm to the base 2. We let RELU be the function $x \mapsto \max\{x,0\}$. For $\mathcal{X} \subset \mathbb{R}$, we denote $\lfloor \mathcal{X} \rfloor := \{\lfloor x \rfloor : x \in \mathcal{X}\}$. For $n \in \mathbb{N}$ and a set $\mathcal{X}$, we denote $\binom{\mathcal{X}}{n} := \{\mathcal{S} \subset \mathcal{X} : |\mathcal{S}| = n\}$ and $[n] := \{0, \ldots, n-1\}$. For $x \geq 0$ and $y > 0$, we denote $x \bmod y := x - y \cdot \lfloor \frac{x}{y} \rfloor$.

Throughout this paper, we consider fully-connected feedforward networks. In particular, we consider the following setup: Given an activation function $\sigma$, we define a neural network $f_\theta$ of $L$ layers (or equivalently $L-1$ hidden layers), input dimension $d_x$, output dimension 1, and hidden layer dimensions $d_1, \ldots, d_{L-1}$ parameterized by $\theta$ as $f_\theta := t_{L-1} \circ \sigma \circ \cdots \circ t_1 \circ \sigma \circ t_0$. Here, $t_\ell : \mathbb{R}^{d_\ell} \to \mathbb{R}^{d_{\ell+1}}$ is an affine transformation parameterized by $\theta$.[4] We denote a neural network using an activation function $\sigma$ by a "$\sigma$ network." We define the width of $f_\theta$ as $\max\{d_1, \ldots, d_{L-1}\}$.

As we introduced in Section 1.1, our main results hold for any sigmoidal activation function and RELU. Here, we formally define the *sigmoidal* functions as follows.

**Definition 2.** *We say a function $\sigma : \mathbb{R} \to \mathbb{R}$ is sigmoidal if the following conditions are satisfied.*

- *Both $\lim_{x \to -\infty} \sigma(x)$, $\lim_{x \to \infty} \sigma(x)$ exist and $\lim_{x \to -\infty} \sigma(x) \neq \lim_{x \to \infty} \sigma(x)$.*

- *There exists $z \in \mathbb{R}$ such that $\sigma$ is continuously differentiable at $z$ and $\sigma'(z) \neq 0$.*

A class of sigmoidal functions covers many activation functions including sigmoid, $\tanh$, hard $\tanh$, etc.[5] Furthermore, since hard $\tanh$ can be represented as a combination of two RELU functions, all results for sigmoidal activation functions hold for RELU as well.[6]

Lastly, we formally define the memorization as follows.

**Definition 3.** *Given $C, d_x \in \mathbb{N}$, a set of inputs $\mathcal{X} \subset \mathbb{R}^{d_x}$, a label function $y : \mathcal{X} \to [C]$, and a neural network $f_\theta : \mathbb{R}^{d_x} \to \mathbb{R}$ parameterized by $\theta$, we say $f_\theta$ can memorize $\{(x, y(x)) : x \in \mathcal{X}\}$ in $d_x$ dimension with $C$ classes if for any $\varepsilon > 0$, there exists $\theta$ such that $|f_\theta(x) - y(x)| \leq \varepsilon$ for all $x \in \mathcal{X}$.*

---

[4]We set $d_0 := d_x$ and $d_L := 1$.

[5]hard $\tanh$ activation function: $x \mapsto -\mathbf{1}[x \leq -1] + x \cdot \mathbf{1}[-1 < x \leq 1] + \mathbf{1}[x > 1]$.

[6]hard $\tanh(x) = \text{RELU}(x+1) - \text{RELU}(x-1) - 1 = \text{RELU}(2 - \text{RELU}(1-x)) - 1$

Definition 3 defines the memorizability as the ability of a network $f_\theta$ to fit a set of input-label pairs. While existing results often define memorization only for the binary labels, we consider arbitrary $C$ classes, and prove our results for general multi-class classification problems. We often write "$f_\theta$ can memorize arbitrary $N$ pairs" without "in $d_x$ dimension with $C$ classes" throughout the paper.

## 4 MAIN RESULTS

### 4.1 MEMORIZATION VIA SUB-LINEAR PARAMETERS

**Efficacy of depth for memorization.** Now, we are ready to introduce our main theorem on memorizing $N$ pairs with $o(N)$ parameters. The proof of Theorem 1 is presented in Section 5.

**Theorem 1.** *For any $C, N, d_x \in \mathbb{N}$, for any $w \in [2/3, 1]$, for any $\Delta \geq 1$, for any sigmoidal activation function $\sigma$, there exists a $\sigma$ network $f_\theta$ of $O\left( \log d_x + \log \Delta + \frac{N^{2-2w}}{1+(1.5w-1)\log N} \log C \right)$ hidden layers and $O\left( d_x + \log \Delta + N^w + N^{1-w/2} \log C \right)$ parameters such that $f_\theta$ can memorize any $\Delta$-separated set of $N$ pairs in $d_x$ dimension with $C$ classes.*

Note that Theorem 1 covers $w = 2/3$ which is not included in its informal version presented in Section 1.1. However, we could only achieve $O(N^{2/3})$ parameters at $w = 2/3$ as the $\log N$ term disappears. In addition, while we only address sigmoidal activation functions in the statement of Theorem 1, note that the same conclusion naturally holds for RELU as we described in Section 3.

In Theorem 1, $\Delta$ only incurs $O(\log \Delta)$ overhead to the number of layers and the number of parameters. As we introduced in Section 1.1, $\log \Delta$ for modern datasets is often very small. Furthermore, $\log \Delta$ is small for random inputs. For example, a set of $d_x$-dimensional i.i.d. standard normal random vectors of size $N$ satisfies $\log \Delta = O(\frac{1}{d_x} \log(N/\sqrt{\delta}))$ with probability at least $1 - \delta$ (see Section C). Hence, the $\Delta$-separateness condition is often negligible.

Suppose that $d_x$ and $C$ are treated as constants, as also assumed in existing results. Then, Theorem 1 implies that if $\log \Delta = O(N^w)$ for some $w < 1$, then $\Theta(N^w)$ (i.e., sub-linear to $N$) parameters are *sufficient* for sigmoidal or RELU networks to memorize arbitrary $\Delta$-separated set of $N$ pairs. Furthermore, if $\log \Delta \leq O(N^{2-2w}/\log N)$ and $w \in (\frac{2}{3}, 1)$, then the network construction in Theorem 1 has $O(N^{2-2w}/\log N)$ layers and $O(N^w)$ parameters. Note that the condition $\log \Delta \leq O(N^{2-2w}/\log N)$ is very loose for many practical datasets, especially for those with huge $N$. Combined with the lower bound $\Omega(N/\log N)$ on the *necessary* number of parameters for RELU networks of constant depth (Bartlett et al., 2019), Theorem 1 implies that depth growing in $N$ is *necessary and sufficient* for memorizing a large class (i.e., $\Delta$-separated) of $N$ pairs with $o(N/\log N)$ parameters. In other words, deeper RELU networks have more memorization power.

**Unimportance of width for memorization.** While depth is critical for the memorization power, we show that the width is not very critical. In particular, we prove that extremely narrow networks of width 3 can memorize with $O(N^{2/3})$ layers (i.e., $O(N^{2/3})$ parameters) as stated in the following theorem. The proof of Theorem 2 is presented in Section F.

**Theorem 2.** *For any $C, N, d_x \in \mathbb{N}$, for any $\Delta \geq 1$, for any sigmoidal activation function $\sigma$, a $\sigma$ network of $\Theta(N^{2/3} \log C)$ hidden layers and width 3 can memorize any $\Delta$-separated set of $N$ pairs in $d_x$ dimension with $C$ classes.*

The statement of Theorem 2 might be somewhat surprising since the network width for memorization does not depend to the input dimension $d_x$. This is in contrast with the recent universal approximation results that width at least $d_x + 1$ is necessary for approximating functions having $d_x$-dimensional domain (Lu et al., 2017; Hanin and Sellke, 2017; Johnson, 2019; Park et al., 2020). The main difference follows from the fundamental difference in the two approximation problems, i.e., approximating a function at finite inputs versus infinite inputs, e.g., the unit cube. Any set of $N$ input vectors can be easily mapped to $N$ "distinct" scalar values by simple projection using inner product. Hence, memorizing finite input-label pairs in $d_x$-dimension can be easily translated into memorizing finite input-label pairs in *one*-dimension. In other words, the dimensionality ($d_x$) of inputs is not very important as long as they can be translated to distinct scalar values. In contrast, there is no "natural" way to map design an injection from $d_x$-dimensional unit cube to lower dimension. Namely, to not loose the "information" from inputs, width $d_x$ for preserving inputs is necessary. Therefore, function approximation in general cannot be done with width independent of $d_x$.

**Extension to regression problem.** The results of Theorem 1 and Theorem 2 can be easily applied to the regression problem, i.e., when labels are from $[0, 1]$. This is because one can simply translate the regression problem with some $\varepsilon > 0$ error tolerance to the classification problem with $\lceil 1/\varepsilon \rceil$ classes. Here, each class $c \in \{0, 1, \ldots, \lceil 1/\varepsilon \rceil\}$ corresponds to the target value $c \cdot \varepsilon$. Hence, the regression problem can also be solved within $\varepsilon$ error with $o(N)$ parameters, where the sufficient number of layers and the sufficient number of parameters are identical to the numbers in Theorem 1 and Theorem 2 with the replacement of $\log C$ with $\log(1/\varepsilon)$.

**Relation with benefits of depth in neural networks.** Our observation that deeper RELU networks have more memorization power is closely related to the recent studies on the benefits of depth in neural networks (Telgarsky, 2016; Eldan and Shamir, 2016; Lin et al., 2017; Arora et al., 2018; Yarotsky, 2018; Chatziafratis et al., 2020a;b). While our observation indicates that depth is critical for the memorization power, these works mostly focused on showing the importance of depth for approximating functions. Here, the existing results on benefits of depth for function approximation cannot directly imply the benefits of depth for memorization since they often focus on specific classes of functions or require parameters far beyond $O(N)$.

## 4.2 GENERIC CRITERIA FOR IDENTIFYING MEMORIZATION POWER

While Theorem 1 proves the existence of networks having $o(N)$ parameters for memorization, the following theorem states the generic criteria for verifying the memorization power of network architectures. The proof of Theorem 3 is presented in Section G.

**Theorem 3.** *For some sigmoidal activation function $\sigma$, let $f_\theta$ be a $\sigma$ network of $L$ hidden layers having $d_\ell$ neurons at the $\ell$-th hidden layer. Then, for any $C, N, d_x \in \mathbb{N}$ and $\Delta \geq 1$, $f_\theta$ can memorize any $\Delta$-separated set of $N$ pairs in $d_x$ dimension with $C$ classes if the following statement holds:*

*There exist $0 < L_1 < \cdots < L_K < L$ for some $2 \leq K \leq \log N$ satisfying conditions 1–4 below.*

1. *$d_\ell \geq 3$ for all $\ell \leq L_K$ and $d_\ell \geq 7$ for all $L_K < \ell$.*

2. *$\prod_{\ell=1}^{L_1} \lfloor (d_\ell + 1)/2 \rfloor \geq \Delta \sqrt{2\pi d_x}$.*

3. *$\sum_{L_{i-1}+1}^{L_i} (d_\ell - 2) \geq 2^{i+3}$ for all $1 < i \leq K - 1$.*

4. *$2^K \cdot \left( \sum_{\ell=L_{K-1}+1}^{L_K} (d_\ell - 2) \right) \cdot \left\lfloor (L - L_K - 1)/\lceil \log C \rceil \right\rfloor - 4 \geq N^2$.*

Our criteria in Theorem 3 require that the layers of the network can be "partitioned" into have $K + 1$ distinct parts characterized by $L_1, \ldots, L_K$ for some $K \geq 2$. Under this partition, we describe the four conditions in Theorem 3 in detail. The first condition suggests that the network width is not very critical. We note that $d_\ell \geq 7$ for $L_K < \ell$ does not contradict Theorem 2 as we highly optimize the network architecture to fit in width 3 for Theorem 2, while we provide generic criteria here.

The second condition considers the first $L_1$ hidden layers. In order to satisfy this condition, deep and narrow architectures are better than shallow and wide architectures under a similar budget for parameters, due to the product form $\prod_{\ell=1}^{L_1} \lfloor (d_\ell + 1)/2 \rfloor$. Nevertheless, the architecture of the first $L_1$ hidden layers is not very critical as only $\log \Delta + \frac{1}{2} \log(2\pi d_x)$ layers are sufficient even with width 3 (e.g., $\log \Delta < 17$ for the ImageNet dataset).

The third condition is closely related to the last condition: As $K$ increases, the LHS in the last condition increases. However, the third condition states that it requires more hidden neurons. Simply, increasing $K$ by one requires doubling hidden neurons from the $(L_1 + 1)$-th hidden layer to the $L_{K-1}$-th hidden layer. Nevertheless, this doubling hidden neurons would make the LHS of the last condition double as well.

The last condition is simple. As we explained, $2^K$ is approximately proportional to the number of hidden neurons from the $(L_1 + 1)$-th hidden layer to the $L_{K-1}$-th hidden layer. The second term in the LHS of the last condition $\sum_{\ell=L_{K-1}+1}^{L_K} (d_\ell - 2)$ is even more simple; it only requires counting hidden neurons. On the other hand, the last term counts the number of layers. This indicates that to satisfy conditions in Theorem 3 using few parameters, the last layers of the network should be deep and narrow. In particular, we note that such a deep and narrow architecture in the last layers is indeed necessary for RELU networks to memorize with $o(N)$ parameters (Bartlett et al., 2019).

Now, we describe how to show memorization with $o(N)$ parameters using our criteria. For simplicity, consider the network with the minimum width stated in the first condition, i.e., $d_\ell = 3$ for all $\ell \le L_K$ and $d_\ell = 7$ for all $L_K < \ell$. As we explained, the second conditions can be easily satisfied. For the third condition, consider choosing $K = \log(N^{2/3})$, then $\Theta(N^{2/3})$ hidden neurons (i.e., $\Theta(N^{2/3})$ hidden layers) would be sufficient, i.e., $L_{K-1} - L_1 = \Theta(N^{2/3})$. Likewise, we choose remaining $L_i$ to satisfy $L_K - L_{K-1} = \Theta(N^{2/3})$ and $L - L_K = \Theta(N^{2/3})$. Then, it naturally satisfy the last condition while using only $\Theta(N^{2/3})$ parameters.

## 5 PROOF OF THEOREM 1

In this section, we give a constructive proof of Theorem 1: We design a $\sigma$ network which can memorize $N$ inputs in $d_x$ dimension with $C$ classes using only $o(N)$ parameters. We note that the proofs of Theorem 2 and Theorem 3 also utilize similar constructions. To construct networks, we first introduce the following lemma, motivated by the RELU networks achieving nearly tight VC-dimension (Bartlett et al., 2019). The proof of Lemma 4 is presented in Section E.1.

**Lemma 4.** *For any $C, V \in \mathbb{N}$ and $p \in [1/2, 1]$, there exists a $\sigma$ network $f_\theta$ of $O\big(\frac{V^{1-p}}{1+(p-0.5)\log V}\big)$ hidden layers and $O(V^p + V^{1/2}\log C)$ parameters such that $f_\theta$ can memorize any $\mathcal{X} \in \binom{\mathbb{R}}{V}$ with $C$ classes satisfying $\lfloor \mathcal{X} \rfloor = [V]$.*

Roughly speaking, Lemma 4 states the following: Suppose that the $V$ inputs are scalar and *well-separated*, in the sense that exactly one input falls into each interval $[i, i+1)$. Then, such well-separated inputs can be mapped to the corresponding labels using only $O(V^{1/2})$ parameters (with $p = 1/2$). Thus, what remains is to build a network mapping $N$ arbitrary inputs to some well-separated set bounded by some $V = o(N^2)$, again using $o(N)$ parameters.

**Projecting input vectors to scalar values.** Now, we introduce our network mapping $\Delta$-separated set of $N$ inputs to some $\mathcal{Z}$ satisfying $\lfloor \mathcal{Z} \rfloor \in \binom{[V]}{N}$. First, we map the input vectors to scalar values using the following lemma. The proof of Lemma 5 is presented in Section E.2.

**Lemma 5.** *For any $\Delta$-separated $\mathcal{X} \in \binom{\mathbb{R}^{d_x}}{N}$, there exist $v \in \mathbb{R}^{d_x}$, $b \in \mathbb{R}$ such that $\lfloor \{v^\top x + b : x \in \mathcal{X}\} \rfloor \in \binom{[O(N^2 \Delta d_x^{1/2})]}{N}$.*

Lemma 5 states that any $\Delta$-separated $N$ vectors can be mapped to some well-separated scalar values bounded by $O(N^2 \Delta d_x^{1/2})$ using a simple projection using $O(d_x)$ parameters. Note that the direct combination of Lemma 4 and Lemma 5 gives us a $\sigma$ network of $O(N\Delta^{1/2}d_x^{1/4})$ parameters which can memorize any $\Delta$-separated $N$ pairs. However, this combination is limited in at least two sense: The required number of parameters (1) has $\Delta^{1/2}d_x^{1/4}$ multiplicative factor and (2) is not sub-linear to $N$. In what follows, we introduce our techniques for resolving these two issues.

**Reducing upper bound to $O(N^2)$.** First, we introduce the following lemma for improving the $\Delta^{1/2}d_x^{1/4}$ multiplicative factor in the number of parameters to $O(\log \Delta + \log d_x)$ additional parameters. The proof of Lemma 6 is presented in Section E.3.

**Lemma 6.** *For any $\mathcal{X} \in \binom{\mathbb{R}}{N}$ such that $\lfloor \mathcal{X} \rfloor \in \binom{[K]}{N}$ for some $K \in \mathbb{N}$, there exists a $\sigma$ network $f$ of $1$ hidden layer and width $3$ such that $\lfloor f(\mathcal{X}) \rfloor \in \binom{[T]}{N}$ where $T := \max\{\lceil K/2 \rceil, \lfloor N^2/4 + 1 \rfloor\}$.*

From Lemma 6, a network of $O(\log \Delta + \log d_x)$ hidden layers and width $3$ can decrease the upper bound $O(N^2 \Delta d_x^{1/2})$ to $\lfloor N^2/4+1 \rfloor$, which enables us to drop the dependence of $\Delta d_x^{1/2}$ on the upper bound using $O(\log \Delta + \log d_x)$ parameters. The intuition behind Lemma 6 is that if the number of target intervals $T$ is large enough compared to the number of inputs $N$, then inputs can be easily mapped without *collision* (i.e., no two inputs are mapped into the same interval) by some simple network of a small number of parameters. For example, we construct $f$ in Lemma 6 as

$$f(x) := \begin{cases} x & \text{if } x \in [0, T) \\ x + b \bmod T & \text{if } x \in [T, K) \end{cases} \tag{1}$$

for some $b \in [T]$. However, if $T$ is not large enough compared to $N$ (i.e., $T < \lfloor N^2/4 + 1 \rfloor$), then our network (1) cannot avoid the collision between inputs, i.e., $b \in [T]$ satisfying $f(\lfloor \mathcal{X} \rfloor) \in \binom{[T]}{N}$

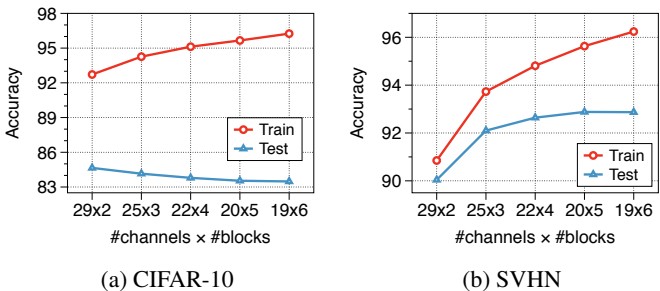

Figure 1: Depth-width trade-off under a similar number of parameters.

may not exist. Hence, combining Lemmas 4–6 only gives us a $\sigma$ network of $O(N + \log \Delta + \log d_x)$ parameters which can memorize any $\Delta$-separated $N$ pairs, i.e., the number of parameters is not sub-linear to $N$.

**Reducing upper bound to** $o(N^2)$**.** To resolve this issue, we further decrease the upper bound using the following lemma. The proof of Lemma 7 is presented in Section E.4.

**Lemma 7.** *For any* $\mathcal{X} \in \binom{\mathbb{R}}{N}$ *such that* $\lfloor \mathcal{X} \rfloor \in \binom{[K]}{N}$ *for some* $K \in \mathbb{N}$*, there exists a* $\sigma$ *network* $f$ *of* 1 *hidden layer and width* $O(N^2/K)$ *such that* $\lfloor f(\mathcal{X}) \rfloor \in \binom{[T]}{N}$ *where* $T := \max \left\{ \lceil K/2 \rceil, N \right\}$*.*

The network in Lemma 7 can reduce the upper bound by approximately half, beyond $\lfloor N^2/4 + 1 \rfloor$; however, the required number of parameters will be doubled if both the current upper bound $K$ decreases by half. Hence, in order to decrease the upper bound from $O(N^2)$ to $V = o(N^2)$ using $\Theta(\log(N^2/V))$ applications of Lemma 7, we need $\Theta(\log(N^2/V))$ layers and $\Theta(N^2/V)$ parameters. Here, we construct each application of Lemma 7 using two hidden layers: one hidden layer of $\Theta(N^2/K)$ hidden neurons for implementing the function and the other hidden layer of one hidden neuron for the output.

**Deriving Theorem 1.** Now, we choose $V = \Theta(N^{2-w})$ for some $w \in [\frac{2}{3}, 1]$. Then, from Lemmas 5–7, $O(\log d_x + \log \Delta + \log N)$ hidden layers and $O(d_x + \log \Delta + N^w)$ parameters suffice for mapping any $\Delta$-separated set of size $N$ to some $\mathcal{Z}$ satisfying $\lfloor \mathcal{Z} \rfloor \in \binom{[V]}{N}$. Finally, from Lemma 4 and choosing $p = \frac{w}{2-w}$, additional $O\left(\frac{N^{2-2w}}{1+(1.5w-1)\log N} \log C\right)$ hidden layers and $O(N^w + N^{1-w/2} \log C)$ parameters suffice for mapping $\mathcal{Z}$ to its labels. This completes the proof of Theorem 1.

## 6 EXPERIMENTS

In this section, we study the effect of depth and width. In particular, we empirically verify whether our theoretical finding extends to practices: Can deep and narrow networks memorize more training pairs than their shallow and wide counterparts under a similar number of parameters? For the experiments, we use residual networks (He et al., 2016) having the same number of channels for each layer. The detailed experimental setups are presented in Section A. In the following experiments, we observe the training and test accuracy of networks by varying the number of channels ($c$) and the number of residual blocks ($b$).

### 6.1 DEPTH-WIDTH TRADE-OFF IN MEMORIZATION

We verify the memorization power of different network architectures having a similar number of parameters. Figure 1 illustrates training and test accuracy of five different architectures with approximately 50000 parameters for classifying the CIFAR-10 dataset (Krizhevsky and Hinton, 2009) and the SVHN dataset (Netzer et al., 2011). One can observe that as the network architecture becomes deeper and narrower, the training accuracy increases. Namely, deep and narrow networks memorize better than shallow and wide networks under a similar number of parameters. This observation agrees with Theorem 1, which states that increasing depth reduces the required number of parameters for memorizing the same number of pairs.

However, more memorization power does not always imply better generalization. In Figure 1b, as the depth increases, the test accuracy also increases for the SVHN dataset. In contrast, the test

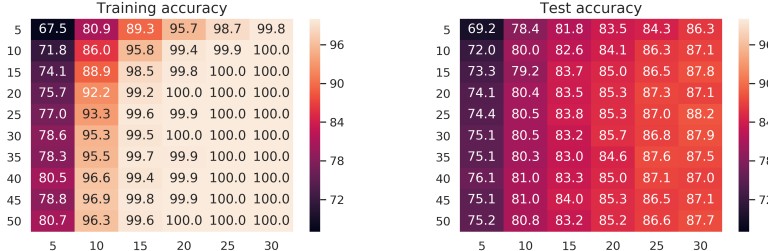

Figure 2: Training and test accuracy by varying width and depth for the CIFAR-10 dataset. The $x$-axis denotes the number of channels and the $y$-axis denotes the number of residual blocks.

accuracy decreases for the CIFAR-10 dataset as the depth increases in Figure 1a. In other words, overfitting occurs for the CIFAR-10 dataset while classifying the SVHN data receive benefits from more expressive power. Note that a similar observation has also been made in the recent double descent phenomenon (Belkin et al., 2019; Nakkiran et al., 2020) that more expressive power can both hurt/improve the generalization.

In addition, this observation can provide guidance on the design of network architectures in applications where the training accuracy and small number of parameters are critical. For example, recent development in video streaming services tries to reduce the traffic by compressing a video by neural networks and send the compressed video and the decoder network (Yeo et al., 2018). Here, the corresponding decoder is often trained only for the video to send; hence, only the training accuracy/loss matters while the decoder's number of parameters should also be small for the traffic.

### 6.2 EFFECT OF WIDTH AND DEPTH

In this section, we observe the effect of depth and width by varying both. Figure 2 reports the training and test accuracy for the CIFAR-10 dataset by varying the number for channels from 5 to 30 and the number of residual blocks from 5 to 50. We present the same experimental results for the SVHN dataset in Section B. First, we observe that a network of 15 channels with feature map size $32 \times 32$ successfully memorize (i.e., over 99% training accuracy). This size is much narrower than modern network architectures, e.g., ResNet-18 has 64 channels at the first hidden layer (He et al., 2016). On the other hand, too narrow network (e.g., 5-channels) fail to memorize. This result does not contradict with Theorem 2 as the test of memorization in experiments/practice involves the stochastic gradient descent. We note that similar phenomenons are observed for the SVHN dataset.

Furthermore, once the network memorize, we observe that increasing width is more effective than increasing depth for improving test accuracy. These results indicate that width is not very critical for the memorization power, while it can be effective for generalization. Note that similar observations have been made in Zhang et al. (2017).

## 7 CONCLUSION

In this paper, we prove that $\Theta(N^{2/3})$ parameters are sufficient for memorizing arbitrary $N$ input-label pairs under the mild $\Delta$-separateness condition. Our result provides significantly improved results, compared to the prior results showing the sufficiency of $\Theta(N)$ parameters with/without conditions on pairs. In addition, Theorem 1 shows that deeper networks have more memorization power. This result coincides with the recent study on the benefits of depth for function approximation. On the other hand, Theorem 2 shows that network width is not important for the memorization power. We also provide generic criteria for identifying the memorization power of networks. Finally, we empirically confirm our theoretical results.

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

# A   EXPERIMENTAL SETUP

In this section, we described the details on residual network architectures and hyper-parameter setups.

We use the residual networks of the following structure. First, a convolutional layer and RELU maps a 3-channel input image to a $c$-channel feature map. Here, the size of the feature map is identical to the size of input images. Then, we apply $b$ residual blocks where each residual block maps $x \mapsto \text{RELU}(\text{CONV} \circ \text{RELU} \circ \text{CONV}(x) + x)$ while preserving the number of channels and the size of feature map. Finally, we apply an average pooling layer and a fully-connected layer. We train the model for $5 \times 10^5$ iterations with batch size $64$ by the stochastic gradient descent. We use the initial learning rate $0.1$, weight decay $10^{-4}$, and the learning rate decay at the $1.5 \times 10^5$-th iteration and the $3.5 \times 10^5$-th iteration by a multiplicative factor $0.1$.

All presented results are averaged over three independent trials.

# B    TRAINING AND TEST ACCURACY FOR SVHN DATASET

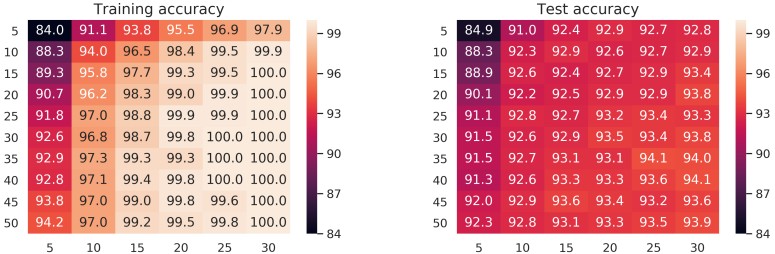

Figure 3: Training and test accuracy by varying width and depth for the SVHN dataset. The $x$-axis denotes the number of channels and the $y$-axis denotes the number of residual blocks.

## C $\Delta$-SEPARATENESS OF GAUSSIAN RANDOM VECTORS

While we mentioned in Section 1.1 that digital nature of data enables the $\Delta$-separateness of inputs with small $\Delta$, random inputs are also $\Delta$-separated with small $\Delta$ with high probability. In particular, we prove the following lemma.

**Lemma 8.** *For any $d_x \in \mathbb{N}$, consider a set of $N$ vectors $\mathcal{X} = \{x_1, \ldots, x_N\} \subset \mathbb{R}^{d_x}$ where each entry of $x_i$ is drawn from the i.i.d. standard normal distribution. Then, for any $\delta > 0$, $\mathcal{X}$ is $\left((N/\sqrt{\delta})^{2/d_x}\sqrt{3e + \frac{5e}{d_x}\ln(N/\sqrt{\delta})}\right)$-separated with probability at least $1 - \delta$.*

Lemma 8 implies that Theorem 1 and Theorem 2 can be successfully applied to random Gaussian input vectors as the $O\left((N/\sqrt{\delta})^{2/d_x}\sqrt{\ln(N/\sqrt{\delta})}\right)$-separateness condition in Lemma 8 is much weaker than our $2^{O(N^{2/3})}$-separateness condition for memorization with $o(N)$ parameters.

*Proof of Lemma 8.* To begin with, we first prove that for $i \neq j$, the following bound holds:

$$
\mathbb{P}\left(\sqrt{\frac{2}{e}} \cdot \left(\frac{N}{\sqrt{\delta}}\right)^{-2/d_x} \leq \frac{1}{\sqrt{d_x}}\|x_i - x_j\|_2 \leq \sqrt{6 + \frac{10}{d_x}\ln\frac{N}{\sqrt{\delta}}}\right)
$$

$$
= \mathbb{P}\left(\frac{2}{e} \cdot \left(\frac{N}{\sqrt{\delta}}\right)^{-4/d_x} \leq \frac{1}{d_x}\|x_i - x_j\|_2^2 \leq 6 + \frac{10}{d_x}\ln\frac{N}{\sqrt{\delta}}\right)
$$

$$
= \mathbb{P}\left(\frac{2}{e} \cdot \left(\frac{N}{\sqrt{\delta}}\right)^{-4/d_x} \leq \frac{2}{d_x}X \leq 6 + \frac{10}{d_x}\ln\frac{N}{\sqrt{\delta}}\right)
$$

$$
= 1 - \mathbb{P}\left(\frac{2}{e} \cdot \left(\frac{N}{\sqrt{\delta}}\right)^{-4/d_x} > \frac{2}{d_x}X\right) - \mathbb{P}\left(\frac{2}{d_x}X > 6 + \frac{10}{d_x}\ln\frac{N}{\sqrt{\delta}}\right)
$$

$$
= 1 - \mathbb{P}\left(\frac{1}{e} \cdot \left(\frac{N}{\sqrt{\delta}}\right)^{-4/d_x} \cdot d_x > X\right) - \mathbb{P}\left(X > \left(3 + \frac{5}{d_x}\ln\frac{N}{\sqrt{\delta}}\right) \cdot d_x\right)
$$

$$
\geq 1 - \left(\frac{1}{e} \cdot \left(\frac{N}{\sqrt{\delta}}\right)^{-4/d_x} e^{1 - \frac{1}{e}\cdot(\frac{N}{\sqrt{\delta}})^{-4/d_x}}\right)^{d_x/2} - \left(\left(3 + \frac{5}{d_x}\ln\frac{N}{\sqrt{\delta}}\right)e^{-2}\left(\frac{N}{\sqrt{\delta}}\right)^{-5/d_x}\right)^{d_x/2}
$$

$$
= 1 - \frac{\delta}{N^2} \cdot \left(e^{-\frac{1}{e}\cdot(\frac{N}{\sqrt{\delta}})^{-4/d_x}}\right)^{d_x/2} - \frac{\delta}{N^2} \cdot \left(\left(3 + \frac{5}{d_x}\ln\frac{N}{\sqrt{\delta}}\right)e^{-2}\left(\frac{N}{\sqrt{\delta}}\right)^{-1/d_x}\right)^{d_x/2}
$$

$$
\geq 1 - \frac{\delta}{N^2} - \frac{\delta}{N^2} \cdot \left(\frac{3}{e^2 N^{1/d_x}} + \frac{5}{e^2} \cdot \frac{\ln\left(\frac{N}{\sqrt{\delta}}\right)^{1/d_x}}{\left(\frac{N}{\sqrt{\delta}}\right)^{1/d_x}}\right)^{d_x/2}
$$

$$
\geq 1 - \frac{\delta}{N^2} - \frac{\delta}{N^2} \cdot \left(\frac{3}{e^2} + \frac{5}{e^3}\right)^{d_x/2}
$$

$$
\geq 1 - \frac{\delta}{N^2} - \frac{\delta}{N^2} = 1 - \frac{2\delta}{N^2} \tag{2}
$$

where $X$ denotes a chi-square random variable with $d_x$ degrees of freedom. For the first inequality in (2), we use the inequalities

$$
\mathbb{P}(X < z \cdot d_x) \leq \inf_{t>0}\frac{\mathbb{E}[e^{-tX}]}{e^{-t(z\cdot d_x)}} = \inf_{t>0}\frac{(1+2t)^{-d_x/2}}{e^{-t(z\cdot d_x)}} = (ze^{1-z})^{d_x/2} \text{ for } 0 < z < 1
$$

$$
\mathbb{P}(X > z \cdot d_x) \leq \inf_{t>0}\frac{\mathbb{E}[e^{tX}]}{e^{t(z\cdot d_x)}} = \inf_{0<t<1/2}\frac{(1-2t)^{-d_x/2}}{e^{t(z\cdot d_x)}} = (ze^{1-z})^{d_x/2} \text{ for } z > 1
$$

which directly follow from the Chernoff bound for the chi-square distribution. For the third inequality in (2), we use the fact that $\max_{x>0}(\ln x)/x = 1/e$ and $N \geq 1$. For the last inequality in (2), we use $\frac{3}{e^2} + \frac{5}{e^3} < 1$.

Then, $\mathcal{X}$ is $\left((N/\sqrt{\delta})^{2/d_x}\sqrt{3e + \frac{5e}{d_x}\ln(N/\sqrt{\delta})}\right)$-separated with probability at least $1 - \delta$ as the following bound holds:

$$\mathbb{P}\left(\sqrt{\frac{2}{e}}\delta^{1/d_x}N^{-2/d_x} \leq \frac{1}{\sqrt{d_x}}\|x_i - x_j\|_2 \leq \sqrt{6 + \frac{10}{d_x}\ln\frac{N}{\sqrt{\delta}}}, \, \forall i \neq j\right)$$

$$= 1 - \mathbb{P}\left(\exists i \neq j \text{ s.t. } \|x_i - x_j\|_2 > \sqrt{6 + \frac{10}{d_x}\ln\frac{N}{\sqrt{\delta}}} \text{ or } \|x_i - x_j\|_2 < \sqrt{\frac{2}{e}}\delta^{1/d_x}N^{-2/d_x}\right)$$

$$\geq 1 - \sum_{i \neq j}\mathbb{P}\left(\|x_i - x_j\|_2 > \sqrt{6 + \frac{10}{d_x}\ln\frac{N}{\sqrt{\delta}}} \text{ or } \|x_i - x_j\|_2 < \sqrt{\frac{2}{e}}\delta^{1/d_x}N^{-2/d_x}\right)$$

$$= 1 - \sum_{i \neq j}\left(1 - \mathbb{P}\left(\sqrt{\frac{2}{e}}\delta^{1/d_x}N^{-2/d_x} \leq \frac{1}{\sqrt{d_x}}\|x_i - x_j\|_2 \leq \sqrt{6 + \frac{10}{d_x}\ln\frac{N}{\sqrt{\delta}}}\right)\right)$$

$$\geq 1 - \frac{N(N-1)}{2} \times \frac{2\delta}{N^2}$$

$$\geq 1 - \delta$$

where the first inequality follows from the union bound and the second inequality follows from (2). This completes the proof of Lemma 8. $\qquad\square$

# D  MAIN PROOF IDEA: MEMORIZATION USING STEP AND ID

## D.1  MAIN IDEA

In the proofs of Theorem 1, Theorem 2, and Theorem 3, we use each sigmoidal activation for approximating either the identity function (ID : $x \mapsto x$) or the binary step function (STEP : $x \mapsto \mathbf{1}[x \geq 0]$). Thus, we construct our network for the proofs using only ID and STEP activation functions, which directly provides the network construction using sigmoidal activation functions (see Section D.2 and Section D.3 for details).

## D.2  TOOLS

We present the following claims important for the proofs. In particular, Claim 9 and Claim 10 guide how to approximate STEP and ID by a single sigmoidal neuron.

**Claim 9** [Kidger and Lyons (2020, Lemma 4.1)]. *For any sigmoidal activation function $\sigma$, for any bounded interval $\mathcal{I} \subset \mathbb{R}$ for any $\varepsilon > 0$, there exist $a, b, c, d \in \mathbb{R}$ such that $|a \cdot \sigma(c \cdot x + d) + b - x| < \varepsilon$ for all $x \in \mathcal{I}$.*

**Claim 10.** *For any sigmoidal activation function $\sigma$, for any $\varepsilon, \delta > 0$, there exist $a, b, c \in \mathbb{R}$ such that $|a \cdot \sigma(c \cdot x) + b - \mathbf{1}[x \geq 0]| < \varepsilon$ for all $x \notin [-\delta, \delta]$.*

*Proof of Claim 10.* We assume that $\alpha := \lim_{x \to -\infty} \sigma(x) < \lim_{x \to \infty} \sigma(x) =: \beta$ where the case that $\beta < \alpha$ can be proved in a similar mannerwdd. From the definition of $\alpha, \beta$, there exists $k > 0$ such that $|\sigma(x) - \alpha| < (\beta - \alpha)\varepsilon$ if $x < -k$ and $|\sigma(x) - \beta| < (\beta - \alpha)\varepsilon$ if $x > k$. Then, choosing $a = \frac{1}{\beta - \alpha}$, $b = -\frac{\alpha}{\beta - \alpha}$, and $c = \frac{k}{\delta}$ completes the proof of Claim 10. $\square$

**Claim 11.** *For any $a, x \in \mathbb{R}$ such that $a \neq 0$, for any $b \in \mathbb{N}$, it holds that $\left\lceil \frac{x}{a \cdot b} \right\rceil = \left\lceil \frac{\lceil x/a \rceil}{b} \right\rceil$.*

*Proof of Claim 11.* It trivially holds that $\left\lceil \frac{x}{a \cdot b} \right\rceil \leq \left\lceil \frac{\lceil x/a \rceil}{b} \right\rceil$. Now, we show a contradiction if $\left\lceil \frac{x}{a \cdot b} \right\rceil < \left\lceil \frac{\lceil x/a \rceil}{b} \right\rceil$. Suppose that $\left\lceil \frac{x}{a \cdot b} \right\rceil < \left\lceil \frac{\lceil x/a \rceil}{b} \right\rceil$. Then, there exists an integer $m$ such that

$$\frac{x}{a \cdot b} \leq m < \frac{\lceil x/a \rceil}{b} \qquad \text{and hence,} \qquad \frac{x}{a} \leq \left\lceil \frac{x}{a} \right\rceil \leq b \cdot m < \left\lceil \frac{x}{a} \right\rceil$$

which leads to a contradiction. This completes the proof of Claim 11. $\square$

## D.3  TRANSFORMING STEP+ID NETWORK TO SIGMOIDAL NETWORK

In this section, we describe how to transform a STEP + ID network into a sigmoidal network within arbitrary error. Formally, we prove the following lemma.

**Lemma 12.** *For any finite set of inputs $\mathcal{X}$, for any STEP + ID network $f$, for any $\varepsilon > 0$, for any sigmoidal activation function $\sigma$, there exists a $\sigma$ network $g$ having the same architecture with $f_\theta$ such that*

$$|f(x) - g(x)| < \varepsilon$$

*for all $x \in \mathcal{X}$.*

Then, the construction in Lemma 12 enables us to construct STEP + ID networks instead of networks using a sigmoidal activation function for proving Theorem 1, Theorem 2, and Theorem 3.

*Proof of Lemma 12.* Without loss of generality, we first assume that for any STEP + ID network $h$, in the evaluation of $h(x)$, all inputs to STEP neurons (i.e., $\mathbf{1}[x \geq 0]$) is non-zero for all $x \in \mathcal{X}$. This assumption can be easily satisfied by adding some small bias to the inputs of STEP neurons where such a bias alwasy exists since $|\mathcal{X}| < \infty$. Furthermore, introducing this assumption does not change $h(x)$ for all $x \in \mathcal{X}$

Now, we describe our construction of $g$. Let $\delta > 0$ be a number satisfying that the absolute value of all inputs to STEP neurons in the evaluation of $f(x)$ for all $x \in \mathcal{X}$ is at least $\delta$. Such $\delta$ always exists due to the assumption we made. Let $L$ be the number of hidden layers in $f$. Starting from $f$, we

iteratively substitute the STEP and ID hidden neurons into the sigmoidal activation $\sigma$, from the last hidden layer to the first hidden layer. In particular, by using Claim 9 and Claim 10, we replace ID and STEP neurons in the $\ell$-th hidden layer by $\sigma$ neurons approximating ID and STEP.

First, let $g_L$ be a network identical to $f$ except for its $L$-th hidden layer consisting of $\sigma$ neurons approximating ID and STEP in $f$. Here, we accurately approximate ID and STEP neurons by $\sigma$ using Claim 9 and Claim 10 so that $|g_L(x) - f(x)| < \varepsilon/L$ for all $x \in \mathcal{X}$. Note that such approximation always exists due to the existence of $\delta > 0$. Now, let $g_{L-1}$ be a network identical to $f_L$ except for its $(L-1)$-th hidden layer consisting of $\sigma$ neurons approximating ID and STEP of $g_L$. Here, we also accurately approximate ID and STEP neurons by $\sigma$ using Claim 9 and Claim 10 so that $|g_L(x) - g_{L-1}(x)| < \varepsilon/L$ for all $x \in \mathcal{X}$. If we repeat this procedure until replacing the first hidden layer, then $g := g_1$ would be the desired network satisfying that $|f(x) - g(x)| < \varepsilon$ for all $x \in \mathcal{X}$. This completes the proof of Lemma 12. □

# E  PROOF OF LEMMAS FOR THEOREM 1

For proving each of Lemma 4–7, we prove stronger technical lemma, as stated in Sections E.1–E.4, and transform the STEP + ID networks to $\sigma$ networks using Lemma 12.

## E.1  PROOF OF LEMMA 4

To prove Lemma 4, we introduce the following lemma. We note that Lemma 13 follows the construction for proving VC-dimension lower bound of RELU networks (Bartlett et al., 2019). The proof of Lemma 13 is presented in Section H.4.

**Lemma 13.** *For any $A, B, D, K, R \in \mathbb{N}$ such that $AB \geq K$, there exists a STEP + ID network $f_\theta$ of $2\lceil \frac{BD}{R} \rceil + 2$ hidden layers and $4A + \left((2R+5)2^R + 2R^2 + 8R + 7\right)\lceil \frac{BD}{R} \rceil - R2^R - R^2 + 3$ parameters satisfying the following property: For any finite set $\mathcal{X} \subset [0, K)$, for any $y : [K] \to [2^D]$, there exists $\theta$ such that $f_\theta(x) = y(\lfloor x \rfloor)$ for all $x \in \mathcal{X}$.*

By choosing $K \leftarrow V$, $A \leftarrow \Theta(V^p)$, $B \leftarrow \Theta(V^{1-p})$, $R \leftarrow 1 + (2p-1)\log V$, and $D \leftarrow \lceil \log C \rceil$ in Lemma 13, there exists a STEP + ID network of $O\left(\frac{V^{1-p}}{1+(p-0.5)\log V}\log C\right)$ hidden layers and $O(V^p)$ parameters which can memorize any $\mathcal{X} \subset \mathbb{R}$ of size $V$ with $C$ classes satisfying $\lfloor \mathcal{X} \rfloor = [V]$. Combining this with Lemma 12 completes the proof of Lemma 4.

## E.2  PROOF OF LEMMA 5

Since the proof is trivial when $d_x = 1$, we consider $d_x \geq 2$. To this end, we first project all $x \in \mathcal{X}$ to $u^\top x \in \mathbb{R}$ by choosing some unit vector $u \in \mathbb{R}^{d_x}$ so that

$$\frac{\max_{\{x,x'\}\in\binom{\mathcal{X}}{2}} |u^\top(x-x')|}{\min_{\{x,x'\}\in\binom{\mathcal{X}}{2}} |u^\top(x-x')|} < N^2 \Delta \sqrt{\frac{\pi d_x}{8}}. \tag{3}$$

Such a unit vector $u$ always exists due to the following lemma. The proof of Lemma 14 is presented in Section H.1.

**Lemma 14.** *For any $N, d_x \in \mathbb{N}$, for any $\mathcal{X} \in \binom{\mathbb{R}^{d_x}}{N}$, there exists a unit vector $u \in \mathbb{R}^{d_x}$ such that $\sqrt{\frac{8}{\pi d_x}}\frac{1}{N^2}\|x-x'\|_2 \leq |u^\top(x-x')| \leq \|x-x'\|_2$ for all $x, x' \in \mathcal{X}$.*

Finally, we construct the desired map by

$$x \mapsto \frac{u^\top x - \min\{u^\top x : x \in \mathcal{X}_1\}}{\min_{\{x,x'\}\in\binom{\mathcal{X}_1}{2}}|u^\top(x-x')|} =: v^\top x + b$$

for some $v \in \mathbb{R}^{d_x}$ and $b \in \mathbb{R}$ so that $\lfloor\{v^\top x + b : x \in \mathcal{X}\}\rfloor \in \binom{\lceil\lceil N^2\Delta\sqrt{\pi d_x/8}\rceil\rceil}{N}$. This completes the proof of Lemma 4.

## E.3  PROOF OF LEMMA 6

To prove Lemma 6, we introduce the following lemma. The proof of Lemma 15 is presented in Section H.2.

**Lemma 15.** *For any $N, K, d \in \mathbb{N}$, for any $\mathcal{X}$ such that $\lfloor \mathcal{X} \rfloor \in \binom{[K]}{N}$, there exists a STEP+ID network $f$ of 1 hidden layer and width $d$ such that $\lfloor f(\mathcal{X}) \rfloor \in \binom{[T]}{N}$ where $T := \max\left\{\lceil\frac{K}{\lfloor(d+1)/2\rfloor}\rceil, \lfloor\frac{N^2}{4}+1\rfloor\right\}$.*

From Lemma 15, one can observe that a STEP + ID network of 1 hidden layer consisting of 3 hidden neuron which can map any $\mathcal{Z}$ such that $\lfloor \mathcal{Z} \rfloor \in \binom{[K]}{N}$ to $\mathcal{Z}'$ such that $\lfloor \mathcal{Z}' \rfloor \in \binom{[T]}{N}$ where $T := \max\left\{\lceil\frac{K}{\lfloor(d+1)/2\rfloor}\rceil, \lfloor\frac{N^2}{4}+1\rfloor\right\}$. Combining Lemma 15 and Lemma 12 completes the proof of Lemma 6.

### E.4 PROOF OF LEMMA 7

In the compression 2 step, we further improve the bound $U := \lfloor \frac{N^2}{4} + 1 \rfloor$ on the hidden feature values to $V = \Theta(N^{2-w})$. Namely, we map $\mathcal{X}_3$ to $\mathcal{X}_4$ such that $\lfloor \mathcal{X}_4 \rfloor \in \binom{[V]}{N}$. To construct such a mapping, we introduce the following lemma. The proof of Lemma 16 is presented in Section H.3.

**Lemma 16.** *For any $N, K, L, d_1, \ldots, d_L \in \mathbb{N}$ such that $N < K$ and $d_\ell \geq 3$ for all $\ell$, for any $\mathcal{X}$ such that $\lfloor \mathcal{X} \rfloor \in \binom{[K]}{N}$, there exists a* STEP $+$ ID *network $f$ of $L$ hidden layers having $d_\ell$ neurons at the $\ell$-th hidden layer such that $\lfloor f(\mathcal{X}) \rfloor \in \binom{[T]}{N}$ where $T := \min \left\{ K, \max \left\{ N \times \lceil \frac{N}{C} \rceil, \lceil \frac{K}{2} \rceil \right\} \right\}$ and $C := \lfloor \frac{1}{2} + \frac{1}{2} \sum_{\ell=1}^{L} (d_\ell - 2) \rfloor$.*

From Lemma 16, a STEP $+$ ID network $f$ of one hidden layer having $\Theta\left(\frac{N^2}{K}\right)$ hidden neurons can map any $\mathcal{X}$ such that $\lfloor \mathcal{X} \rfloor \in \binom{[K]}{N}$ to $f(\mathcal{X})$ such that $\lfloor f(\mathcal{X}) \rfloor = \binom{[T]}{N}$ where $T := \max\{N, \lceil \frac{K}{2} \rceil\}$. Combining this and Lemma 12 completes the proof of Lemma 7.

# F   Proof of Theorem 2

In this proof, we construct a STEP + ID network and transform it to a $\sigma$ network using Lemma 12, as in the proof of Theorem 1 in Section 5 and Section E. In particular, Theorem 2 is a direct consequence of Lemma 14, Lemma 15, Lemma 16, and Lemma 17 presented as follows. The proof Lemma 17 is presented in Section H.7.

**Lemma 17.** *For any $A, B, D, K \in \mathbb{N}$ such that $AB \geq K$, there exists a STEP + ID network $f_\theta$ of $A + (2D + 1)B$ hidden layers and width 3 satisfying the following property: For any finite set $\mathcal{X} \in [0, K)$, for any $y : [K] \to [2^D]$, there exists $\theta$ such that $f_\theta(x) = y(\lfloor x \rfloor)$.*

From Lemma 14, Lemma 15, and Claim 11, a STEP + ID network of $\lceil \log(\Delta\sqrt{2\pi d_x}) \rceil$ hidden layers and width 3 can map a $\Delta$-separated set of inputs $\mathcal{X}_1$ to $\mathcal{X}_2$ such that $\lfloor \mathcal{X}_2 \rfloor \in \binom{[U]}{N}$ where $U := \lfloor \frac{N^2}{4} + 1 \rfloor$. From Lemma 16, a STEP+ID network of $\sum_{i=1}^{\lceil \log(U/V) \rceil - 1} \left( \lceil \frac{2N}{\lfloor U/(2^i N) \rfloor} \rceil - 1 \right) + \lceil \frac{2N}{\lfloor V/N \rfloor} \rceil - 1$ hidden layers and width 3 can map $\mathcal{X}_2$ to $\mathcal{X}_3$ such that $\lfloor \mathcal{X}_3 \rfloor \in \binom{[V]}{N}$ for some $N \leq V \leq U$. Here, we will choose $V = \Theta(N^{4/3})$. Finally, from Lemma 17, for any $A, B \in \mathbb{N}$ such that $AB \geq V$, a STEP + ID network of $A + (2D + 1)B$ hidden layers and width 3 can map $\mathcal{X}_3$ to their labels where $D := \lceil \log C \rceil$. We will choose $A, B = \Theta(N^{2/3})$ to satisfy $AB \geq V = \Theta(N^{4/3})$.

Hence, for any $A, B, V \in \mathbb{N}$ such that $N \leq V \leq U$ and $AB \geq V$, a STEP + ID network of $\lceil \log(\Delta\sqrt{2\pi d_x}) \rceil + \sum_{i=1}^{\lceil \log(U/V) \rceil - 1} \left( \lceil \frac{2N}{\lfloor U/(2^i N) \rfloor} \rceil - 1 \right) + \lceil \frac{2N}{\lfloor V/N \rfloor} \rceil + A + (2D + 1)B - 1$ hidden layers and width 3 can memorize arbitrary $\Delta$-separated set of size $N$. Note that combining functions does not require additional hidden layer as the linear maps constructing the outputs of functions can be absorbed into the first linear map in the next function.

Finally, substituting $V \leftarrow \lceil N^{4/3} \rceil$, $A \leftarrow \lceil \sqrt{(2D+1)V} \rceil$, and $B \leftarrow \lceil \sqrt{V/(2D+1)} \rceil$ and using Lemma 12 result in the statement of Theorem 2. This completes the proof of Theorem 2.

## G  PROOF OF THEOREM 3

The proof of Theorem 3 have the same structure of the proof of Theorem 1 consisting of four steps: projection, compression 1, compression 2, and learning. In particular, we construct a STEP + ID network and use Lemma 12 as in the proofs of Theorem 1 and Theorem 2.

For the network construction, we divide the function of $f_\theta$ into four disjoint parts, as in Section 5. The first part does not utilize a hidden layers but construct project input vectors into scalar values. The second part corresponds to the first $L_1$ hidden layers decreases the upper bound on scalar values to $O(N^2)$. The third part corresponds to the next $L_{K-1} - L_1$ hidden layers further decreases the upper bound to $o(N^2)$. The last part corresponds to the rest hidden layers construct a network mapping hidden features to their labels.

Now, we describe our construction in detail. To begin with, let us denote a $\Delta$-separated set of inputs by $\mathcal{X}_1$. First, from Lemma 14, one can project $\mathcal{X}_1$ to $\mathcal{X}_2$ such that $\lfloor \mathcal{X}_2 \rfloor \in \binom{\lceil \lceil N^2 \Delta \sqrt{\pi d_x/8} \rceil \rceil}{N}$. Note that the projection step does not require to use hidden layers as it can be absorbed into the linear map before the next hidden layer. Then, from Lemma 15, the first $L_1$ hidden layers can map $\mathcal{X}_2$ to $\mathcal{X}_3$ such that $\lfloor \mathcal{X}_3 \rfloor \in \binom{\lceil \lfloor N^2/4+1 \rfloor \rceil}{N}$ since

$$\prod_{\ell=1}^{L_1} \left\lfloor \frac{d_\ell + 1}{2} \right\rfloor \geq \Delta\sqrt{2\pi d_x} \Rightarrow \prod_{\ell=1}^{L_1} \left\lfloor \frac{d_\ell + 1}{2} \right\rfloor \geq \frac{N^2 \Delta\sqrt{\pi d_x/8}}{N^2/4}$$

$$\Rightarrow \frac{N^2}{4} \geq \frac{N^2 \Delta\sqrt{\pi d_x/8}}{\prod_{\ell=1}^{L_1} \lfloor (d_\ell + 1)/2 \rfloor}$$

$$\Rightarrow \left\lfloor \frac{N^2}{4} + 1 \right\rfloor \geq \frac{N^2 \Delta\sqrt{\pi d_x/8}}{\prod_{\ell=1}^{L_1} \lfloor (d_\ell + 1)/2 \rfloor}$$

$$\Rightarrow \left\lfloor \frac{N^2}{4} + 1 \right\rfloor \geq \left\lceil \frac{N^2 \Delta\sqrt{\pi d_x/8}}{\prod_{\ell=1}^{L_1} \lfloor (d_\ell + 1)/2 \rfloor} \right\rceil.$$

Note that we also utilize Claim 11 for the sequential application of Lemma 15. Consecutively, from Lemma 16, the next $L_{K-1} - L_1$ hidden layers can map $\mathcal{X}_3$ to $\mathcal{X}_4$ such that $\lfloor \mathcal{X}_4 \rfloor \in \binom{\lceil \lceil U/2^{K-2} \rceil \rceil}{N}$ since the third condition holds and

$$\sum_{\ell=L_{i-1}+1}^{L_i} (d_\ell - 2) \geq 2^{i+3} \Rightarrow \frac{1}{2} \sum_{\ell=L_{i-1}+1}^{L_i} (d_\ell - 2) \geq 2^{i+2}$$

$$\Rightarrow \left\lfloor \frac{1}{2} + \frac{1}{2} \sum_{\ell=L_{i-1}+1}^{L_i} (d_\ell - 2) \right\rfloor \geq 2^{i+2}$$

$$\Rightarrow \frac{N}{\left\lfloor \frac{1}{2} + \frac{1}{2}\sum_{\ell=L_{i-1}+1}^{L_i}(d_\ell - 2) \right\rfloor} \leq \frac{N}{2^{i+2}}$$

$$\Rightarrow \left\lceil \frac{N}{\left\lfloor \frac{1}{2} + \frac{1}{2}\sum_{\ell=L_{i-1}+1}^{L_i}(d_\ell - 2) \right\rfloor} \right\rceil \leq \frac{N}{2^{i+1}}$$

$$\Rightarrow N \times \left\lceil \frac{N}{\left\lfloor \frac{1}{2} + \frac{1}{2}\sum_{\ell=L_{i-1}+1}^{L_i}(d_\ell - 2) \right\rfloor} \right\rceil \leq \frac{N^2/4}{2^{i-1}}$$

$$\Rightarrow N \times \left\lceil \frac{N}{\left\lfloor \frac{1}{2} + \frac{1}{2}\sum_{\ell=L_{i-1}+1}^{L_i}(d_\ell - 2) \right\rfloor} \right\rceil \leq \frac{\lfloor N^2/4 + 1 \rfloor}{2^{i-1}}$$

$$\Rightarrow N \times \left\lceil \frac{N}{\left\lfloor \frac{1}{2} + \frac{1}{2}\sum_{\ell=L_{i-1}+1}^{L_i}(d_\ell - 2) \right\rfloor} \right\rceil \leq \left\lceil \frac{\lfloor N^2/4 + 1 \rfloor}{2^{i-1}} \right\rceil$$

where we use the inequality $\lceil a \rceil \leq 2a$ for $a \geq 1/2$ and the assumption $K \leq \log N$, i.e., $N/2^{i+1} \geq 1$ for $i \leq \log N - 1$. Here, we also utilize Claim 11 for the sequential application of Lemma 16.

Now, we reinterpret the last condition as follows.

$$2^K \cdot \left( \sum_{\ell=L_{K-1}+1}^{L_K} (d_\ell - 2) \right) \cdot \left\lfloor (L - L_K - 1)/\lceil \log C \rceil \right\rfloor \geq N^2 - 4$$

$$\Rightarrow \left( \sum_{\ell=L_{K-1}+1}^{L_K} (d_\ell - 2) \right) \cdot \left\lfloor (L - L_K - 1)/\lceil \log C \rceil \right\rfloor \geq \frac{N^2/4 + 1}{2^{K-2}}$$

$$\Rightarrow \left( \sum_{\ell=L_{K-1}+1}^{L_K} (d_\ell - 2) \right) \cdot \left\lfloor (L - L_K - 1)/\lceil \log C \rceil \right\rfloor \geq \left\lceil \frac{\lfloor N^2/4 + 1 \rfloor}{2^{K-2}} \right\rceil$$

Finally, using the following lemma and the above inequality, the rest hidden layers can map $\mathcal{X}_4$ to their corresponding labels (by choosing $L' := L_3$). The proof of Lemma 18 is presented in Section H.8.

**Lemma 18.** *For $D, K, L, d_1, \ldots, d_L \in \mathbb{N}$, suppose that there exist $0 < L' < L$ and $r_{L'+1}, \ldots, r_{L-1} \in \mathbb{N}$ satisfying that for $r_{L'} = r_L = 1$*

$$\left( \sum_{\ell=1}^{L'} (d_\ell - 2) \right) \cdot \left\lfloor \frac{\sum_{\ell=L'+1}^{L-1} r_\ell}{D} \right\rfloor \geq K \quad \text{and} \quad 2^{r_\ell} + r_\ell + r_{\ell-1} + 3 \leq d_\ell \text{ for all } L' + 1 \leq \ell \leq L.$$

*Then, there exists a STEP + ID network $f_\theta$ of $L$ hidden layers having $d_\ell$ hidden neurons at the $\ell$-th hidden layer such that for any finite $\mathcal{X} \subset [0, K)$, for any $y : [K] \to [2^D]$, there exists $\theta$ satisfying $f_\theta(x) = y(\lfloor x \rfloor)$ for all $x \in \mathcal{X}$.*

Choose $K \leftarrow \lceil U/2^{K-2} \rceil$ and $r_\ell = 1$ for all $\ell$ in Lemma 18 completes our construction of the STEP + ID network. Note that we utilize the condition that $d_\ell \geq 7$ for all $L_K < \ell$ here. Using Lemma 12 completes the proof of Theorem 3.

# H PROOFS OF TECHNICAL LEMMAS

## H.1 PROOF OF LEMMA 14

Since the upper bound holds for any unit vector $u \in \mathbb{R}^{d_x}$, we focus on the lower bound. In addition, if $N = 1$, the result of Lemma 14 trivially holds. Hence, we assume $N \geq 2$, i.e., $\sqrt{\frac{8}{\pi d_x}} \frac{1}{N^2} < 1$. In this proof, we show that given any vector $v \in \mathbb{R}^{d_x}$ such that $\|v\|_2 \neq 0$, a random unit vector $u \in \mathbb{R}^{d_x}$ from the uniform distribution satisfies that

$$\mathbb{P}\left( \frac{|u^\top v|}{\|v\|_2} < \sqrt{\frac{8}{\pi d_x}} \frac{1}{N^2} \right) < \frac{2}{N^2}. \tag{4}$$

This implies that there exists a unit vector $u \in \mathbb{R}^{d_x}$ such that $\sqrt{\frac{8}{\pi d_x}} \frac{1}{N^2} \|x - x'\|_2 \leq |u^\top(x - x')|$ for all $x, x' \in \mathcal{X}$, due to the following union bound: for $\mathcal{V} = \left\{ x - x' : \{x, x'\} \in \binom{\mathcal{X}}{2}, x \leq x' \right\}$ for some total order $\leq$ on $\mathcal{X}$,

$$\mathbb{P}\left( \bigcup_{v \in \mathcal{V}} \left\{ \frac{|u^\top v|}{\|v\|_2} < \sqrt{\frac{8}{\pi d_x}} \frac{1}{N^2} \right\} \right) \leq \sum_{v \in \mathcal{V}} \mathbb{P}\left( \frac{|u^\top v|}{\|v\|_2} < \sqrt{\frac{8}{\pi d_x}} \frac{1}{N^2} \right) < \frac{N(N-1)}{2} \times \frac{2}{N^2} < 1.$$

Now we prove (4). To begin with, we show that the following equality holds for any $v \in \mathbb{R}$:

$$\mathbb{P}\left( \frac{|u^\top v|}{\|v\|_2} < \sqrt{\frac{8}{\pi d_x}} \frac{1}{N^2} \right) = \mathbb{P}\left( |u_1| < \sqrt{\frac{8}{\pi d_x}} \frac{1}{N^2} \right).$$

Here, the equality follows from choosing $v/\|v\|_2 = (1, 0, \ldots, 0)$ using the symmetry. Furthermore, $\mathbb{P}\left( |u_1| < \sqrt{\frac{8}{\pi d_x}} \frac{1}{N^2} \right)$ can be bounded by

$$\mathbb{P}\left( |u_1| < \sqrt{\frac{8}{\pi d_x}} \frac{1}{N^2} \right) = \mathbb{P}\left( 0 < u_1 < \sqrt{\frac{8}{\pi d_x}} \frac{1}{N^2} \right) + \mathbb{P}\left( -\sqrt{\frac{8}{\pi d_x}} \frac{1}{N^2} < u_1 \leq 0 \right)$$

$$= 2 \times \mathbb{P}\left( 0 < u_1 < \sqrt{\frac{8}{\pi d_x}} \frac{1}{N^2} \right)$$

$$= \frac{2}{\texttt{Area}_{d_x}(1)} \times \int_{\arccos\left(\sqrt{\frac{8}{\pi d_x}} \frac{1}{N^2}\right)}^{\frac{\pi}{2}} \texttt{Area}_{d_x - 1}(\sin\phi) d\phi$$

$$= 2 \times \frac{\texttt{Area}_{d_x - 1}(1)}{\texttt{Area}_{d_x}(1)} \times \int_{\arccos\left(\sqrt{\frac{8}{\pi d_x}} \frac{1}{N^2}\right)}^{\frac{\pi}{2}} \sin^{d_x - 2}\phi d\phi$$

$$= 2 \times \frac{2\pi^{\frac{d_x - 1}{2}}/\Gamma(\frac{d_x - 1}{2})}{2\pi^{\frac{d_x}{2}}/\Gamma(\frac{d_x}{2})} \times \int_{\arccos\left(\sqrt{\frac{8}{\pi d_x}} \frac{1}{N^2}\right)}^{\frac{\pi}{2}} \sin^{d_x - 2}\phi d\phi$$

$$< 2 \times \sqrt{\frac{d_x}{2\pi}} \times \int_{\arccos\left(\sqrt{\frac{8}{\pi d_x}} \frac{1}{N^2}\right)}^{\frac{\pi}{2}} 1 d\phi$$

$$= \sqrt{\frac{2d_x}{\pi}} \left( \frac{\pi}{2} - \arccos\left( \sqrt{\frac{8}{\pi d_x}} \frac{1}{N^2} \right) \right)$$

$$= \sqrt{\frac{2d_x}{\pi}} \arcsin\left( \sqrt{\frac{8}{\pi d_x}} \frac{1}{N^2} \right)$$

$$\leq \sqrt{\frac{2d_x}{\pi}} \times \frac{\pi}{2} \times \sqrt{\frac{8}{\pi d_x}} \frac{1}{N^2} = \frac{2}{N^2}$$

where $\texttt{Area}_d(r) := 2\pi^{\frac{d}{2}} r^{d-1}/\Gamma\left(\frac{d}{2}\right)$ denotes the surface area of a hypersphere of radius $r$ in $\mathbb{R}^d$ and $\Gamma(x)$ denotes the gamma function. Here, the second equality follows from the symmetry and $\mathbb{P}(u_1 = 0) = 0$. The first inequality follows from $\sin\phi \leq 1$ and $\frac{\Gamma\left(\frac{d_x}{2}\right)}{\Gamma\left(\frac{d_x - 1}{2}\right)} < \sqrt{\frac{d_x}{2}}$ from the Gautschi's inequality (see Lemma 19). The second inequality follows from $\phi \leq \frac{\pi}{2} \sin\phi$ for $0 \leq \phi \leq \frac{\pi}{2}$. This completes the proof of Lemma 14.

**Lemma 19** [Gautschi's inequality (Gautschi, 1959)]. *For any $x > 0$, for any $s \in (0, 1)$,*

$$x^{1-s} < \frac{\Gamma(x+1)}{\Gamma(x+s)} < (x+1)^{1-s}.$$

## H.2   PROOF OF LEMMA 15

In this proof, we assume that $K > N^2/4$ and $T = \left\lceil \frac{K}{\lfloor (d+1)/2 \rfloor} \right\rceil$ where other cases trivially follow from this case. To begin with, we first define a network $f_b(x) : [0, K) \to [0, T)$ for $b = (b_i)_{i=1}^{\lfloor (d-1)/2 \rfloor} \in [T]^{\lfloor (d-1)/2 \rfloor}$ as

$$
f_b(x) := \begin{cases} x & \text{if } x < T \\ x + b_1 \bmod T & \text{if } T \leq x < 2T \\ x + b_2 \bmod T & \text{if } 2T \leq x < 3T \\ \qquad \vdots \\ x + b_{\lfloor (d-1)/2 \rfloor} \bmod T & \text{if } \lfloor \frac{d-1}{2} \rfloor T \leq x \end{cases}
$$

$$
= x - T \times \mathbf{1}[x \geq T] + \sum_{i=1}^{\lfloor (d-1)/2 \rfloor} \left( \left( b_i - \sum_{j=1}^{i-1} b_j \right) \times \mathbf{1}[x \geq iT] - T \times \mathbf{1}[x + b_i \geq (i+1)T] \right).
$$

(5)

One can easily observe that $f_b$ can be implemented by a STEP + ID network of 1 hidden layer and width $d$ as $T \times \mathbf{1}[x \geq T]$ in (5) can be absorbed into $\left( b_i - \sum_{j=1}^{i-1} b_j \right) \times \mathbf{1}[x \geq iT]$ in (5) for $i = 1$.

Now, we show that if $T > N^2/4$, then there exist $b \in [T]^{\lfloor (d-1)/2 \rfloor}$ such that $\left| \lfloor f_b(\mathcal{X}) \rfloor \right| = N$ to complete the proof. Our proof utilizes the mathematical induction on $i$: If there exist $b_1, \ldots, b_{i-1} \in [T]$ such that

$$
\lfloor f_b(\{x \in \mathcal{X} : x < iT\}) \rfloor = \binom{[T]}{|\lfloor \{x \in \mathcal{X} : x < iT\} \rfloor|},
$$

(6)

then there exists $b_i \in [T]$ such that

$$
\lfloor f_b(\{x \in \mathcal{X} : x < (i+1)T\}) \rfloor = \binom{[T]}{|\lfloor \{x \in \mathcal{X} : x < (i+1)T\} \rfloor|}.
$$

(7)

Here, one can observe that the statement trivially holds for the base case, i.e., for $\{x \in \mathcal{X} : x < T\}$. Now, using the induction hypothesis, suppose that there exist $b_1, \ldots, b_{i-1} \in [T]$ satisfying (6). Now, we prove that there exists $b_i \in [T]$ such that $\mathcal{S}_{b_i} := \lfloor f_b(\{x \in \mathcal{X} : iT \leq x < (i+1)T\}) \rfloor$ does not intersect with $\mathcal{T} := \lfloor f_b(\{x \in \mathcal{X} : x < iT\}) \rfloor$, i.e, (7) holds. Consider the following inequality:

$$
\sum_{b_i \in [T]} |\mathcal{S}_{b_i} \cap \mathcal{T}| = |\mathcal{S}_{b_i}| \times |\mathcal{T}| \leq \frac{N^2}{4}
$$

where the equality follows from the fact that for each $x \in \{x \in \mathcal{X} : iT \leq x < (i+1)T\}$, there exists exactly $|\mathcal{T}|$ values of $b_i$ so that $\lfloor x \bmod T \rfloor \in \mathcal{T}$. However, since the number of possible choices of $b_i$ is $T$, if $T > N^2/4$, then there exists $b_i \in [T]$ such that $\mathcal{S}_{b_i} \cap \mathcal{T} = \emptyset$, i.e., (7) holds. This completes the proof of Lemma 15.

### H.3 Proof of Lemma 16

The proof of Lemma 16 is similar to that of Lemma 15. To begin with, we first define a network $f_b(x) : [0, K] \to [0, T)$ for $b = (b_i)_{i=1}^C \in [T]^C$ and $T =: M_1 < M_2 < \cdots < M_{C+1} := K$ so that $|\{x \in \mathcal{X} : M_i \le x < M_{i+1}\}| \le \lceil \frac{N}{C} \rceil \le \lfloor \frac{T}{N} \rfloor$ for all $i$ as

$$
f_b(x) := \begin{cases} x & \text{if } x < T \\ x + b_1 \bmod T & \text{if } M_1 = T \le x < M_2 \\ x + b_2 \bmod T & \text{if } M_2 \le x < M_3 \\ \quad\vdots \\ x + b_C \bmod T & \text{if } M_C \le x \end{cases}
$$

$$
= x - 2T \times \mathbf{1}[x \ge T] + \sum_{i=1}^C \left( \left(T + b_i - \sum_{j=1}^{i-1} b_j\right) \times \mathbf{1}[x \ge M_i] \right.
$$

$$
\left. - T \times \mathbf{1}\big[x \ge \min\{2T - b_i, M_{i+1}\}\big] \right) \tag{8}
$$

where (8) holds as $T \ge \lceil \frac{K}{2} \rceil$. Here, one can easily implement $f_b$ by a STEP + ID network of $L$ hidden layer and $d_\ell$ neurons at the $\ell$-th hidden layer by utilizing one neuron for storing the input $x$, another one neuron for storing the temporary output, and other neurons implement indicator functions at each layer. This is because one do not require to store $x$ in the last hidden layer and there exists $2C$ indicator functions to implement ($T \times \mathbf{1}[x \ge T]$ in (8) can be absorbed into $\left(T + b_i - \sum_{j=1}^{i-1} b_j\right) \times \mathbf{1}[x \ge M_i]$ in (8) for $i = 1$).

Now, we show that if $T \ge N$, then there exist $b \in [T]^C$ such that $\big|\lfloor f_b(\mathcal{X}) \rfloor\big| = N$ to completes the proof. Our proof utilizes the mathematical induction on $i$: If there exist $b_1, \dots, b_{i-1} \in [T]$ such that

$$
\lfloor f_b(\{x \in \mathcal{X} : x < M_i\}) \rfloor = \binom{[T]}{\big|\lfloor \{x \in \mathcal{X} : x < M_i\} \rfloor\big|}, \tag{9}
$$

then there exists $b_i \in [T]$ such that

$$
\lfloor f_b(\{x \in \mathcal{X} : x < M_{i+1}\}) \rfloor = \binom{[T]}{\big|\lfloor \{x \in \mathcal{X} : x < M_{i+1}\} \rfloor\big|}. \tag{10}
$$

Here, one can observe that the statement trivially holds for the base case, i.e., for $\{x \in \mathcal{X} : x < T\}$. From the induction hypothesis, suppose that there exist $b_1, \dots, b_{i-1} \in [T]$ satisfying (9). Now, we prove that there exists $b_i \in [T]$ such that $\mathcal{S}_{b_i} := \lfloor f_b(\{x \in \mathcal{X} : M_i \le x < M_{i+1}\}) \rfloor$ does not intersect with $\mathcal{T} := \lfloor f_b(\{x \in \mathcal{X} : x \le M_i\}) \rfloor$, i.e., (10) holds. Consider the following inequality:

$$
\sum_{b_i \in [T]} |\mathcal{S}_{b_i} \cap \mathcal{T}| \le \lfloor \tfrac{T}{N} \rfloor \times \left(N - \lfloor \tfrac{T}{N} \rfloor\right) < T
$$

where the first inequality follows from the fact that $|\mathcal{S}_{b_i}| \le \lfloor \frac{T}{N} \rfloor$, $|\mathcal{T}| \le \left(N - \lfloor \frac{T}{N} \rfloor\right)$, and for each $x \in \{x \in \mathcal{X} : M_i \le x < M_{i+1}\}$, there exists exactly $|\mathcal{T}|$ values of $b_i$ so that $\lfloor x \bmod T \rfloor \in \mathcal{T}$. However, since the number of possible choices of $b_i$ is $T$, there exists $b_i \in [T]$ such that $\mathcal{S}_{b_i} \cap \mathcal{T} = \emptyset$, i.e, (10) holds. This completes the proof of Lemma 16.

### H.4    PROOF OF LEMMA 13

In this proof, we explicitly construct $f_\theta$ satisfying the desired property stated in Lemma 13. To begin with, we describe the high-level idea of the construction. First, we construct a map $g : x \mapsto \left(\lfloor \frac{x}{B} \rfloor, x \bmod B\right)$ to transforming an input $x \in [0, K)$ to a pair $(a, b)$ such that $a \in [A]$ and $\lfloor b \rfloor \in [B]$. Here, we give labels to the pair $(a, b)$ corresponding to the input $x$ as $y(a, \lfloor b \rfloor) := y(\lfloor x \rfloor)$. Note that this label is well-defined as if $\lfloor x_1 \rfloor \neq \lfloor x_2 \rfloor$, then $\lfloor g(x_1) \rfloor \neq \lfloor g(x_2) \rfloor$. Now, we construct parameters $w_0, \ldots, w_{A-1}$ containing the label information of $\mathcal{X}$ as

$$w_a := \sum_{c \in [B]} y(a, c) \times 2^{-(c+1)D}, \tag{11}$$

i.e., from the $(\lfloor b \rfloor \cdot D + 1)$-th bit to the $(\lfloor b \rfloor \cdot D + D)$-th bit of the $a$-th parameter $(w_a)$ contains the label information of $(a, \lfloor b \rfloor)$. Under this construction, we recover the label of $x \in \mathcal{X}$ by first mapping $x$ to the pair $(a, b)$ and extracting the $z$-th parameter $w_a$. Then, using STEP, we extract bits from the $(\lfloor b \rfloor \cdot D + 1)$-th bit to the $(\lfloor b \rfloor \cdot D + D)$-th bit of $w_a$ to recover the label.

Now, we explicitly construct the above procedure. To this end, we introduce the following lemma. The proof of Lemma 20 is presented in Section H.5.

**Lemma 20.** *For any $A, B, K, L, d_1, \ldots d_L \in \mathbb{N}$ such that $AB \geq K$ and $d_\ell \geq 2$ for all $\ell$, for any $w_0, \ldots, w_{A-1} \in \mathbb{R}$, for any finite set $\mathcal{X} \subset [0, K)$, if $\sum_{\ell=1}^{L}(d_\ell - 2) \geq A$, then there exists a STEP+ID network $f$ of $L$ layers and $d_\ell$ neurons at the $\ell$-th layer such that $f(x) = (w_{\lfloor x/B \rfloor}, x \bmod B)$ for all $x \in \mathcal{X}$.*

From Lemma 20, a STEP + ID network of 1 hidden layer consisting of $A + 2$ hidden neurons can map $x \in \mathcal{X}$ to $(w_{\lfloor x/B \rfloor}, x \bmod B)$. Note that this network requires overall $4A + 10$ parameters ($3A + 6$ edges and $A + 4$ biases).

Finally, we introduce the following lemma for extracting from the $(\lfloor b \rfloor \cdot D + 1)$-th bit to the $(\lfloor b \rfloor \cdot D + D)$-th bit of $w_a$. The proof of Lemma 21 is presented in Section H.6.

**Lemma 21.** *For any $D, B, R \in \mathbb{N}$, for any finite set $\mathcal{X} \subset [0, B)$, there exists a STEP + ID network $f$ of $2\lceil \frac{BD}{R} \rceil$ hidden layers and $\left((2R + 5)2^R + 2R^2 + 8R + 7\right)\lceil \frac{BD}{R} \rceil - R2^R - R^2 + 3$ parameters satisfying the following property: For any $w = \sum_{i=1}^{BD} u_i \times 2^{-i}$ such that $u_i \in \{0, 1\}$, $f(x, w) = \sum_{i=1}^{D} u_{\lfloor x \rfloor \cdot D + i} \times 2^{D-i}$ for all $x \in \mathcal{X}$.*

From Lemma 21, a STEP + ID network of $2\lceil \frac{BD}{R} \rceil$ hidden layers and $\left((2R + 5)2^R + 2R^2 + 8R + 7\right)\lceil \frac{BD}{R} \rceil - R2^R - R^2 + 3$ parameters. Hence, by combining Lemma 20 and Lemma 21, $f_\theta$ can be implemented by a STEP + ID network of $2\lceil \frac{BD}{R} \rceil + 2$ hidden layers and $4A + \left((2R + 5)2^R + 2R^2 + 8R + 7\right)\lceil \frac{BD}{R} \rceil - R2^R - R^2 + 3$ parameters. This completes the proof of Lemma 13.

## H.5 PROOF OF LEMMA 20

We design $f$ as $f := f_L \circ \cdots \circ f_1(0, x)$ where each $f_\ell$ represents the function of the $\ell$-th layer consisting of $d_\ell$ neurons. In particular, we construct $f_\ell$ as follows:

$$f_\ell(w, x) := \left( w + w_0 \times \mathbf{1}[\ell = 1] + \sum_{i=1}^{d_\ell - 2} (w_{c_\ell + i} - w_{c_\ell + i - 1}) \times \mathbf{1}[x \geq iB], \right.$$
$$\left. x - \sum_{i=1}^{d_\ell - 2} B \times \mathbf{1}[x \geq iB] \right)$$

where $w_{-1} := 0$ and $c_\ell := \sum_{i=1}^{\ell-1}(d_\ell - 2)$. Then, $f$ is the desired function and each $f_\ell$ can be implemented by a STEP + ID networks of 1 hidden layer consisting of $d_\ell$ hidden neurons (two neurons for storing $x, w$ and other neurons are for $d_\ell - 2$ indicator functions). This completes the proof of Lemma 20.

## H.6 PROOF OF LEMMA 21

We construct $f(x, w) := 2^{R\lceil BD/R\rceil + D} \times f_{\lceil BD/R\rceil} \circ \cdots \circ f_1(x, w)$ where $f_\ell$ is defined as

$$
f_\ell(x, v) := \begin{cases}
\left(x, v - \sum_{i=1}^{R} u_{(\ell-1)R+i} \times 2^{-(\ell-1)R-i} \right. \\
\quad \left. + \sum_{i=1}^{R} \left(u_{(\ell-1)R+i} \wedge \mathbf{1}[m_{i,\ell} \leq x < m_{i,\ell}+1]\right) \times 2^{-r_{i,\ell}}\right) & \text{if } \ell < \lceil \frac{BD}{R} \rceil \\
v - \sum_{i=1}^{R} u_{(\ell-1)R+i} \times 2^{-(\ell-1)R-i} \\
\quad + \sum_{i=1}^{R} \left(u_{(\ell-1)R+i} \wedge \mathbf{1}[m_{i,\ell} \leq x < m_{i,\ell}+1]\right) \times 2^{-r_{i,\ell}} & \text{if } \ell = \lceil \frac{BD}{R} \rceil
\end{cases}.
$$

where $u_i$ denotes the $i$-th bit of $w$ in the binary representation, $\wedge$ denotes the binary 'and' operation, and $m_{i,\ell}, r_{i,\ell}$ are defined as

$$
m_{i,\ell} := \left\lfloor \frac{(\ell-1)R+i}{D} \right\rfloor
$$

$$
r_{i,\ell} := \left\lceil \frac{BD}{R} \right\rceil R + \left((\ell-1)R+i-1 \bmod D\right) + 1.
$$

Namely, each $f_\ell$ extracts $R$ bits from the input $w$ and it store the extracted bits to the last bits of $v$ if the extracted bits are in from $(\lfloor x \rfloor \cdot D + 1)$-th bit to the $(\lfloor x \rfloor \cdot D + D)$-th bit of $w$. Thus, $f(x, w)$ is the desired function for Lemma 21.

To implement each $f_\ell$ by a STEP + ID network, we introduce Lemma 22. Note that we extract $u_i$ from $w$ in Lemma 22, i.e., we do not assume that $u_i$ is given. From Lemma 22, a STEP + ID network of $2\lceil \frac{BD}{R} \rceil$ hidden layers consisting of $2^R + R + 1$ and $R + 2$ hidden neurons alternatively can map $(x, w)$ to $\sum_{i=1}^{D} u_{\lfloor x \rfloor \cdot D+i} \times 2^{D-i}$ for all $x \in \mathcal{X}$. By considering the input dimension 2 and the output dimension 1, this network requires $\left((2R+5)2^R + 2R^2 + 8R + 7\right)\lceil \frac{BD}{R} \rceil - R2^R - R^2 + 3$ parameters $\left(\left((2R+4)2^R + 2R^2 + 6R + 4\right)\lceil \frac{BD}{R} \rceil - R2^R - R^2 + 2$ edges and $\left(2^R + 2R + 3\right)\lceil \frac{BD}{R} \rceil + 1$ biases$\right)$. This completes the proof of Lemma 21.

**Lemma 22.** *A* STEP + ID *network of 2 hidden layers having $2^R + R + 1$ and $R + 2$ hidden neurons at the first and the second hidden layer, respectively, can implement $f_\ell$.*

*Proof of Lemma 22.* We construct $f_\ell := g_3 \circ (g_2 \oplus g_1)$ where $g_2 \oplus g_1$ denotes the function concatenating the outputs of $g_1, g_2$. In this proof, we mainly focus on constructing $f_\ell$ for $\ell < \lceil \frac{BD}{R} \rceil$ since $f_{\lceil BD/R\rceil}$ can be implemented similarly. We define $g_1, g_2, g_3$ as

$$
g_1(x, v) := \left( x, v, \sum_{i=0}^{2^R-1} \eta_{1,i} \times \mathbf{1}[i \times 2^{-\ell R} \leq v < (i+1) \times 2^{-\ell R}], \right.
$$

$$
\vdots,
$$

$$
\left. \sum_{i=0}^{2^R-1} \eta_{R,i} \times \mathbf{1}[i \times 2^{-\ell R} \leq v < (i+1) \times 2^{-\ell R}] \right)
$$

$$
= (x, v, u_{(\ell-1)R+1}, \ldots, u_{\ell R})
$$

$$
g_2(x, v) := \left( \mathbf{1}[m_{1,\ell} \leq x < m_{1,\ell}+1], \ldots, \mathbf{1}[m_{R,\ell} \leq x < m_{R,\ell}+1] \right)
$$

$$
g_3 \circ (g_1 \oplus g_2) := \left( x, v - \sum_{i=1}^{R} u_{(\ell-1)R+i} \times 2^{-(\ell-1)R-i} \right.
$$

$$
\left. + \sum_{i=1}^{R} \mathbf{1}\left[u_{(\ell-1)R+i} + \mathbf{1}[m_{i,\ell} \leq x < m_{i,\ell}+1] \geq 2\right] \times 2^{-r_{i,\ell}} \right)
$$

$$
= \left( x, v - \sum_{i=1}^{R} u_{(\ell-1)R+i} \times 2^{-(\ell-1)R-i} \right.
$$

$$
\left. + \sum_{i=1}^{R} \left(u_{(\ell-1)R+i} \wedge \mathbf{1}[m_{i,\ell} \leq x < m_{i,\ell}+1]\right) \times 2^{-r_{i,\ell}} \right).
$$

where $\eta_{r,i}$ is a constant such that $\eta_{r,i} = 1$ if $i \times 2^{-\ell R} \le x < (i+1) \times 2^{-\ell R}$ implies that the $((\ell-1)R+r)$-th bit of $x$ is 1 and $\eta_{r,i} = 0$ otherwise. Here, one can easily observe that $g_1$ can be implemented by a linear combinations of $\mathbf{1}[v \ge 2^{-\ell R}], \ldots, \mathbf{1}[v \ge (2^R - 1) \times 2^{-\ell R}]$ as it trivially holds that $\mathbf{1}[v \ge 0]$ and $\mathbf{1}[v < 2^{-(\ell-1)R}]$, i.e., $2^R - 1$ indicator functions are enough for $g_1$. Hence, $g_1$ can be implemented by a STEP + ID network of 1 hidden layer consisting of $2^R + 1$ hidden neurons where additional 2 neurons are for passing $x, v$. In addition, $g_2$ can be implemented by a STEP + ID network of $R$ hidden neurons. Finally, $g_3$ can be implemented by a STEP + ID network of 1 hidden layer consisting of $R + 2$ hidden neurons ($R$ neurons for $R$ indicator functions and 2 neurons for passing $x, v$).

Therefore, $f_\ell$ can be implemented by a STEP + ID network of 2 hidden layers consisting of $2^R + R + 1$ hidden neurons for the first hidden layer and $R + 2$ hidden neurons for the second hidden layer. Note that implementation within two hidden layer is possible since the outputs of $g_1, g_2$ are simply linear combination of their hidden activation values and hence, can be absorbed into the linear map between hidden layers. This completes the proof of Lemma 22. $\qquad\square$

## H.7    PROOF OF LEMMA 17

The main idea of the proof of Lemma 17 is identical to that of Lemma 13. Recall $A, B$ and $w_0, \ldots, w_{A-1} \in \mathbb{R}$ from the proof of Lemma 13. From Lemma 20, for any finite set $\mathcal{X} \subset [0, K)$, for any $w_0, \ldots, w_{A-1} \in \mathbb{R}$, a STEP + ID network of $A$ hidden layers and width 3 can map $x$ to $(w_{\lfloor x/B \rfloor}, x \bmod B)$ for all $x \in \mathcal{X}$. Now, we introduce the following lemma replacing Lemma 21.

**Lemma 23.** *For any $D, B \in \mathbb{N}$, for any finite set $\mathcal{X} \subset [0, B)$, for any $w = \sum_{i=1}^{DB} u_i \times 2^{-i}$ for some $u_i \in \{0, 1\}$, there exists a STEP + ID network $f$ of $(2D + 1)B$ hidden layers and width 3 such that $f(x, w) = \sum_{i=1}^{D} u_{\lfloor x \rfloor \cdot D + i} \times 2^{-i}$ for all $x \in \mathcal{X}$.*

Using Lemma 20 and Lemma 23, one can easily find a STEP + ID network of $A + (2D+1)B$ hidden layers and width 3 satisfying the condition in Lemma 17. This completes the proof of Lemma 17.

*Proof of Lemma 23.* We construct $f(x, w) := 2^{D(B+1)} \times f_{DB} \circ g_{DB} \circ h_{DB} \circ \cdots \circ f_1 \circ g_1 \circ h_1$ where $f_\ell, g_\ell, h_\ell$ are defined as

$$h_\ell(x, w) := \begin{cases} (x, w) & \text{if } \ell \bmod D \neq 1 \\ (x - 1 + 3K \times \mathbf{1}[x < 0], w) & \text{if } \ell \bmod D = 1 \end{cases}$$

$$g_\ell(x, w) := \left(x, w - 2^{-\ell} \times \mathbf{1}[w \geq 2^{-\ell}], \mathbf{1}[w \geq 2^{-\ell}]\right)$$

$$f_\ell(x, w, n) = \left(x, w + 2^{-DK - d(\ell)} \times \mathbf{1}[x - n < -1]\right)$$

where $d(\ell) := \ell - \lfloor \frac{\ell-1}{D} \rfloor + 1$. Now, we explain the constructions of $f_\ell, g_\ell, h_\ell$. Let $b \in [0, B)$ and $v \in [0, 1)$ be inputs of $f$, i.e., consider $f(b, v)$. First, the indicator function in $h_\ell$ is activated only at $\ell = (\lfloor b \rfloor + 1) \cdot D + 1$ as $b < B$. In particular, the first entry of the output of $h_{(\lfloor b \rfloor + 1) \cdot D + 1}$ is greater than $2B$ and this is the maximum value of the first entry of the output of $h_{(\lfloor b \rfloor + 1) \cdot D + 1}$ as it monotonically decreases as $\ell$ grows. The indicator function in $g_\ell$ extracts and outputs the $\ell$-th bit of $u$. Lastly, $f_\ell$ add the $\ell$-th bit of $u$ if and only if $\ell \in \{\lfloor b \rfloor \cdot D + 1, \ldots, (\lfloor b \rfloor + 1) \cdot D\}$. This is because $x \in [-1, 0)$ if and only if $\ell \in \{\lfloor b \rfloor \cdot D + 1, \ldots, (\lfloor b \rfloor + 1) \cdot D\}$.

Here, $h_\ell$ at $\ell \bmod D = 1$ can be implemented by a STEP + ID network of 1 hidden layer and width 3, $g_\ell$ can be implemented by a STEP + ID network of 1 hidden layer and width 3, and $f_\ell$ can be implemented by a STEP + ID network of 1 hidden layer and width 3. Hence, $f$ can be implemented by a STEP + ID network of $(2D + 1)B$ hidden layers and width 3. This completes the proof of Lemma 23. $\qquad\square$

## H.8   PROOF OF LEMMA 18

The proof of Lemma 18 is almost identical to the proof of Lemma 13. Recall $A, B$ and $w_0, \ldots, w_{A-1} \in \mathbb{R}$ as in the proof of Lemma 13. From Lemma 24, for any finite set $\mathcal{X} \subset [0, K)$, for any $w_0, \ldots, w_{A-1} \in \mathbb{R}$, the first $L'$ hidden layers of the STEP + ID network $f_\theta$ can map $x$ to $(w_{\lfloor x/B \rfloor}, x \bmod B)$ for all $x \in \mathcal{X}$. Now, we introduce the following lemma replacing Lemma 21. Using Lemma 24 completes the proof of Lemma 18.

**Lemma 24.** *For any $D, B, R, L, d_1, \ldots, d_L \in \mathbb{N}$, for any finite set $\mathcal{X} \subset [0, B)$, suppose that there exists $r_1, \ldots, r_{L-1} \in \mathbb{N}$ satisfying that for $r_0 = r_L = 1$,*

$$\sum_{\ell=1}^{L-1} r_\ell \geq BD,$$
$$2^{r_\ell} + r_\ell + r_{\ell-1} + 3 \leq d_\ell \qquad \text{for all} \qquad 1 \leq \ell \leq L.$$

*Then, there exists a STEP + ID network $f$ of $L$ hidden layers having $d_\ell$ hidden neurons at the $\ell$-th hidden layer satisfying the following property: For any $w = \sum_{i=1}^{BD} u_i \times 2^{-i}$ such that $u_i \in \{0, 1\}$, $f(x, w) = \sum_{i=1}^{D} u_{\lfloor x \rfloor \cdot D + i} \times 2^{D-i}$ for all $x \in \mathcal{X}$.*

*Proof of Lemma 24.* The proof of Lemma 24 utilizes the network constructions in the proofs of Lemma 21 and Lemma 22. In particular, we construct $f(x, w) := 2^{D + \sum_{\ell=1}^{L-1} r_\ell} \times f_{L-1} \circ \cdots \circ f_1(x, w)$ defined as

$$f_\ell(x, v) := \begin{cases} \left(x, v - \sum_{i=1}^{r_\ell} u_{R_\ell+i} \times 2^{-R_\ell-i} \right. \\ \qquad \left. + \sum_{i=1}^{r_\ell} \left(u_{R_\ell+i} \wedge \mathbf{1}[m_{i,\ell} \leq x < m_{i,\ell}+1]\right) \times 2^{-s_{i,\ell}} \right) & \text{if } \ell < \lceil \frac{BD}{R} \rceil \\ v - \sum_{i=1}^{r_\ell} u_{R_\ell+i} \times 2^{-R_\ell-i} \\ \qquad + \sum_{i=1}^{r_\ell} \left(u_{R_\ell+i} \wedge \mathbf{1}[m_{i,\ell} \leq x < m_{i,\ell}+1]\right) \times 2^{-s_{i,\ell}} & \text{if } \ell = \lceil \frac{BD}{R} \rceil \end{cases}.$$

where $u_i$ denotes the $i$-th bit of $w$ in the binary representation, $\wedge$ denotes the binary 'and' operation, and $R_\ell, m_{i,\ell}, s_{i,\ell}$ are defined as

$$R_\ell := \sum_{i=1}^{\ell-1} r_\ell,$$
$$m_{i,\ell} := \left\lfloor \frac{R_\ell + i}{D} \right\rfloor,$$
$$s_{i,\ell} := R_L + (R_\ell + i - 1 \bmod D) + 1.$$

Note that $R_1 = 0$ as the summation starts from $i = 1$ and $R_L = \sum_{\ell=1}^{L-1} r_\ell$. Namely, each $f_\ell$ extracts $r_\ell$ bits from the input $w$ and it store the extracted bits to the last bits of $v$ if the extracted bits are in from $(\lfloor x \rfloor \cdot D + 1)$-th bit to the $(\lfloor x \rfloor \cdot D + D)$-th bit of $w$. Thus, $f(x, w)$ is the desired function for Lemma 24.

We construct $f_\ell(x, v)$ using the $\ell$-th hidden layer and the $(\ell+1)$-th hidden layer, i.e., there exists an overlap between constructions of $f_\ell(x, v)$ and $f_{\ell+1}(x, v)$. In particular, Lemma 22 directly allows us to obtain such a construction under the assumption in Lemma 18 that

$$2^{r_\ell} + r_\ell + r_{\ell-1} + 3 \leq d_\ell \qquad \text{for all} \qquad 1 \leq \ell \leq L.$$

This completes the proof of Lemma 24. □

