# OpenReview forum: "Provable Memorization via Deep Neural Networks using Sub-linear Parameters"
_ICLR.cc/2021/Conference — Reject_

### Official Review · AnonReviewer3 · 2020-10-26
**Review for Provable Memorization via Deep Neural Networks using Sub-linear Parameters**

**Rating:** 5
**Confidence:** 4

**Review:**

This paper proves that $\Theta(N^{2/3})$ parameters are sufficient for memorizing arbitrary N input-label pairs under the mild $\Delta$-separation condition. Overall, it is an interesting theoretical paper and obtains an improved rate over previous works. There are some concerns given as follows:
1. The $\Delta$-separated set seems to be an important condition. The paper claims that $\log \Delta$ is not a big number. I am wondering, when the data points are iid from some distribution such as a multivariate normal distribution, what is $\Delta$ in terms of sample size? In this case, $\Delta$ will depends on both the sample size and the input dimension.
2. The theoretical developments in this paper are for fully connected feedforward networks. How to extend it to CNNs? The experiments are for ResNets. There seems to have a gap.
3.The paper considers the network where the output dimension is 1. How to extend this to networks with more than 1-dimensional output? It is not clear to me in the Definition 3, how to use a network with 1-d output to memorize data with C classes.
4. What is the connection among memorization, training/testing accuracy, and generalization? In the experiment part, the authors sometimes try to show there are some connection but sometimes try to show there are no relationship. It is confusing.
5. Can authors highlight the novelty of the proof? Are there any new techniques beyond those from previous papers?

---

> ### Author Response · Authors · 2020-11-22
> **Response to Reviewer 3**
>
> We sincerely appreciate your valuable comments, efforts, and time. We updated the draft according to the reviewers' comments and colored update parts blue. We address each of your comments in detail as follows.
>
> **$\Delta$-separateness of i.i.d. Gaussian.** We proved that for a random set $\mathcal X=\{x_1,\dots,x_N\}\subset\mathbb R^{d_x}$ where each entry of $x_i$ is sampled from i.i.d. standard normal distribution, $\mathcal X$ is $O\Big((N/\sqrt{\delta})^{2/d_x}\sqrt{\ln (N/\sqrt{\delta})}\Big)$-separated with probability at least $1-\delta$. We note that this separateness result easily satisfies our $2^{O(N^{2/3})}$-separateness condition for memorization with a sub-linear number of parameters, even at the worst case: $d_x=1$ (recall that larger $\Delta$ in $\Delta$-separateness means milder assumption). We added the detailed derivation of this result in Section C in the revised draft.
>
>
> **Memorization with $C$ classes.** We first note that we define memorization with $C$ classes in Definition 3 by providing scalar-valued labels in $\{0,1\dots,C-1\}$ to inputs, i.e., the labels are 1-d for multi-class classification tasks.
> Nevertheless, our results memorizing inputs with multi-class scalar-valued labels can be easily generalized to one-hot labels by additionally introducing a network which maps $\{0,1\dots,C-1\}$ to corresponding one-hot labels. We hope our answer clarifies your concern.
>
>
> **Connection among memorization, training/test accuracy, and generalization.** We first describe the connection between memorization power and training accuracy. Under the assumption that optimization methods (e.g., SGD) can find a global minimum, the training accuracy can fully characterize how many input-label pairs in the training dataset can be mapped by a given network. Hence, the training accuracy is closely related to the memorization power of networks. In experiments, we focused on verifying whether our theoretical finding also holds in practice: Can deep and narrow architectures memorize more data in the training dataset than shallow and wide counterparts under a similar number of parameters? As illustrated in Figure 1, we find that it is true under our experimental setup.
>
> Our experiments also investigated the connection between memorization power (training accuracy) and generalization (test accuracy) in experiments, but it seems that there is no clear correlation between the two. Figure 1 illustrates that better memorization power does not always imply better generalization. We remarked in the experiment section that similar observations have been made in the recent papers on the double descent phenomenon that more memorization power can both hurt/improve generalization.
>
> We clarified these points in the experiment section in the revised draft.
>
> **Using convolutional layers and residual connections for experiments.** Although our results are for fully-connected networks, convolutional networks are more actively used in practice. Therefore, we performed experiments on convolutional networks as we focused on verifying whether our theoretical finding (e.g., deep and narrow networks can memorize more than shallow and wide counterparts) also holds in practice.
>
> We use residual connections for the experiments since, without residual connections, SGD often converges to a local optimum of a deep network whose training accuracy is lower than that of shallower networks of the same width. Namely, the training accuracy can capture memorization power without residual connections. Hence, to avoid this issue, introducing residual connections was inevitable.

---

> > ### Author Response · Authors · 2020-11-22
> > **Response to Reviewer 3**
> >
> > **Novelty of proof.**
> > Our proof framework has the following novel characteristics.
> >
> > - Our network construction mimics the feature extraction of real networks. Specifically, our construction first extracts some 'general features' from the inputs regardless of their labels. Here, our general features are chosen to be 'well-separated' in the sense that they can be easily mapped to 'any' labels. In particular, as the hidden layers of real networks gradually improve the quality of hidden features, the hidden layers of our construction are designed to gradually improve the quality of the 'well-separateness' of our general features. We note that extracting these general features within a sub-linear number of parameters is the core of our proof (see Section 5 in the revised draft for more details). After extracting the general features, the rest part of our construction maps the general feature to their labels. In contrast, many existing memorization literature focused on clustering inputs of the same label without extracting general features.
> >
> > - Our proof framework extracts 'scalar' hidden features for preserving the information from an input vector, while many existing works maintain 'vector' hidden features. As preserving/passing the scalar hidden features requires only one hidden node, it enables us to construct the network within a very narrow width (Theorem 2).
> >
> > We added Section 5 for the proof of our main theorem in the revised draft.
> >
> > **Extension to CNNs.** As our network construction for the proof of our results does not have symmetric weights, we believe that it is non-trivial to extend our results to convolutional networks directly. However, we believe that proving that a sub-linear number of parameters are sufficient for memorization beyond the fully-connected networks would be an interesting future research direction.

---

### Official Review · AnonReviewer2 · 2020-10-28
**Recommendation to accept**

**Rating:** 6
**Confidence:** 3

**Review:**

Summary:

This paper makes the following four contributions.

Firstly, it is shown (Theorem 1) that \Theta(N^{2/3}) parameters are sufficient for neural networks --with sigmoidal activation functions -- to memorize N input-label pairs, under an admittedly mild “\Delta-separated” assumption on the input points (Definition 1). This is an improvement over existing results which show that \Theta(N) parameters suffice, albeit typically for arbitrary N input-label pairs. As discussed in the paper, this result implies that depth is crucial for memorization with sub-linear parameters.

Secondly, the authors show (Theorem 2) that for fully connected networks of width 3, with sigmoidal activations,  \Theta(N^{2/3} + log(\Delta)) parameters suffice for memorizing any \Delta-separated set of N input-label pairs. This implies that the network width does not necessarily have to increase with N for memorization with sub-linear number of parameters.

Thirdly, the authors study the question of identifying the maximum number of input-label pairs a given network can memorize and provide general criteria (Theorem 3) for the same. Existing results for this problem show that the number of arbitrary input-label pairs that can be memorized is at most linear in the number of parameters. The result in this paper shows that memorization of \Delta separated points can be done with the number of pairs super linear in the number of parameters.

Finally, empirical evidence is provided which support their theoretical results, namely that deep networks memorize better than shallow networks (With the same number of parameters).


----------------------------------------------------------------------------
Reasons for score:

The paper provides novel theoretical results for the problem of memorization via deep neural networks. After the discussion phase, however, the significance of these results is a bit unclear to me.

------------------------------------------------------------------------------
Pros:

-	Very well written and clearly structured paper. The problem has been motivated nicely in the introduction.
-	The related work section is rigorous and clearly details the current results for this problem.
-	The empirical results are convincing and support the theoretical claims in the paper.

-------------------------------------------------------------------------------
Cons:

No proof sketch is provided within the main text for any of the theorems. Studying finite sample expressivity of a network in an infinite precision setting also seems a bit strange (See "post discussion" below). Some minor remarks (typos etc.) are listed below, I am hoping that the authors will provide the clarifications stated therein in the rebuttal phase.

------------------------------------------------------------------------------
Minor comments:
-	Typo on pg. 3 (Section 2.1): d_{max} and d_{max} --> d_{max} and d_{min}
-	Typo on pg. 5: Definition 3 define the memorizability … --> Definition 3 defines memorizability …
-	On pg. 2, 3rd para after Theorem 1: I think the opening sentence is true provided
log(\Delta) is less than N^{w}, isn’t it?
-	Towards the bottom of pg. 2, after the Theorem of Bartlett et al., it is mentioned that Theorem 1 together with Bartlett’s theorem imply that depth necessarily has to increase with N for memorization with sublinear number of parameters (this is stated on pg. 5 as well). This implication was not clear to me, some additional explanation will be helpful.
-	The discussion after theorem 2 (starting on pg. 5) involving the comparison with the function approximation problem is a bit unclear, especially the part on pg. 6 which discusses transforming the d_x dimensional inputs to “distinct” scalar values.

-----------------------------------------------------------------------------
Post discussion:

Following the discussion phase, the significance of these results seems to be a bit unclear to me. For instance, suppose that we are allowed to construct the sigmoidal activation. Theorem 2.1 in  the following paper https://hal.archives-ouvertes.fr/hal-01256489/document  states that for any continuous + univariate function $f$, and $\epsilon > 0$, there exists a sigmoidal activation function in $C^{\infty}$ and an associated neural network with one neuron in the hidden layer (with the aforementioned sigmoidal activation) such that $f$ can be uniformly approximated to accuracy $\epsilon$ by this neural network. Therefore the set of N points in $\mathbb{R}^{d_X}$ can be first mapped to distinct points on a line, and then be  shattered/memorized by the aforementioned one-neuron neural network. Of course, the setup in the present paper considers the sigmoidal activation to be fixed (e.g., ReLU) and this is a non-trivial difference. But still, it is not clear just how big of a difference this is.

---

> ### Author Response · Authors · 2020-11-22
> **Response to Reviewer 2**
>
> We sincerely appreciate your valuable comments, efforts, and time. We updated the draft according to the reviewers' comments and colored updated parts blue. We address each of your comments in detail as follows.
>
> **Proof sketch.** According to the reviewer's comment, we added Section 5 for the proof of our main theorem (Theorem 1) in the revised draft.
>
> **Opening sentence in the third paragraph after Theorem 1.** We thank the reviewer for pointing this out.
> What the reviewer expected is correct and we clarified this point in the revised draft.
>
> **Implication of Bartlett et al. and our results.** We thank the reviewer for pointing this out. The result by Bartlett et al. shows that for ReLU networks with a constant depth (i.e., depth does not increase with $N$), $\Theta(N/\log N)$ parameters are necessary for memorizing at least one set of $N$ inputs with arbitrary labels. On the other hand, when depth grows with $N$, our theorems show the sufficiency of $o(N/\log N)$ parameters for memorizing a large class (i.e., $\Delta$-separated) of $N$ pairs.
> Combining these two results indicate that for memorizing a large class of $N$ pairs with $o(N/\log N)$ parameters, depth growing with $N$ is necessary (from Bartlett et al.) and sufficient (from our results). We clarified this point in the revised draft.
>
> **Comparison with the necessary width for function approximation.** The discrepancy between the necessary/sufficient widths for the memorization (width 3 is sufficient) and the universal approximation (width $d_x+1$ is necessary) is from the nature of mapping finite inputs and infinite inputs (e.g., all inputs in the unit cube) to their labels. For a simple illustration, suppose we have a set of $d_x$-dimensional $N$ distinct input vectors $x_1, \dots, x_N$. If we choose a $d_x$-dimensional unit vector $u$ uniformly at random and take inner products $u^T x_i$, then the scalar values $u^T x_1, \dots, u^T x_N$ are distinct with probability 1. Thus, $N$ distinct vectors can be easily mapped to $N$ distinct scalar values. As a result, memorizing finite pairs in $d_x$-dimension can be translated into memorizing finite pairs in one-dimension. In other words, the dimensionality of inputs is not critical and hence, the sufficient size of width for the memorization does not depend on $d_x$.
>
> On the other hand, for the function approximation, mapping all infinite inputs (e.g., on the unit cube) in a $d_x$-dimensional space to distinct lower-dimensional vectors is highly non-trivial, e.g., there is no continuous bijection between $[0,1]^2$ and $[0,1]$. Hence, width at least $d_x$ is necessary for preserving the input information in $d_x$-dimensional space, to prevent input vectors with different outputs from being mapped to the same lower-dimensional vector.
>
> We clarified these points in the revised draft.
>
> **Typos.** We thank the reviewer for finding typos. We fixed them in the revised draft.

---

### Official Review · AnonReviewer1 · 2020-10-29

**Rating:** 7
**Confidence:** 4

**Review:**

==== Summary ====

The paper studies the memorization capacity of deep networks as a function of the number of parameters. Many prior works have shown that to memorize $N$ examples $O(N)$ parameters are sufficient and that to memorize any set of $N$ examples $\Omega(N)$ parameters are necessary. This work shows that under very mild and commonly satisfied conditions, $O(N^{\frac{2}{3}})$ parameters and layers are sufficient for memorizing $N$ examples, a significant improvement over prior results. Additionally, they show that even very narrow width-bounded networks can memorize with sub-linear parameters, but the same is not true for depth-bounded networks, demonstrating a new capability unlocked by deeper networks. Finally, they characterize the properties sufficient for memorizing $N$ examples by a given network.


==== Detailed Review ====

Main strengths:
* A significant improvement over prior bounds. Memorization with sub-linear parameters impacts our understanding of how networks work and has practical implications.
* Presents a novel perspective on the benefits of depth, showing that it is critical for optimal utilization of parameters for memorization.

Main weaknesses:
* The proof method is not discussed in the main body. The proof sketch that is in the appendix provides very little insight and is hard to digest.

I recommend acceptance in light of the drastic reduction in the bound and its implications, as well as the novel perspective on the role of depth. The authors mention large supervised datasets as a motivation, but my impression is that the real implications are for self-supervised learning, where datasets are 100 times larger. For instance, GPT-3, a 170B-parameters model, was trained to predict roughly 240B unique tokens. Results prior to this suggested that even with this enormous model, it would be hard to overfit all the examples. In contrast, this new result indicates that a model 5x smaller could memorize this huge dataset completely.

Despite the above, while I'm in favor of accepting the paper, I gave it only a score of 6 because of the lack of discussion and motivation of the proof techniques used in the appendices. The proof sketch in the appendix is too dense and technical, making it hard to decipher without going into the details of the actual proofs. For example, just by looking at the sketch, there seems to be no difference between the first and second compression stages. While it is clear the need for the first stage (dropping dependency on $\Delta$), going from $U$ to $V$ seems meaningless because $V$ barely characterized beyond being between $N$ and $U$. I will be more than happy to raise my score once the authors update the paper with a clearer proof sketch and discussion of key proof methods in the body of the paper.

As a minor note, it seems as though everywhere where $\Theta$ is used, it should be replaced with big-O notation instead, as most of these instances are upper-bounds and not tight bounds on the minimal number of parameters sufficient to achieve memorization.

==== Update following rebuttal and discussion ====

The AC has raised an important point that the paper does not discuss its reliance on infinite precision. While using infinite precision is not necessarily wrong, it does make the paper's result less relevant in practice, narrowing its contributions to being mostly about theoretical aspects. More importantly, by hiding this technical detail deep within the proofs, the authors missed the opportunity to discuss the difference between the number of bits used by the parameters and the number of edges (operations) necessary for memorization. It seems to me that the number of bits is linear with the number of examples, even as you can reduce the number of edges, which does have practical implications about the utilization of the memory bought by depth. Moreover, if the authors had discussed this topic, it would've been accepted more easily.

Despite the above flaws, I still find value in this article, even if its contributions are mostly theoretical.

---

> ### Author Response · Authors · 2020-11-22
> **Response to Reviewer 1**
>
> We sincerely appreciate your valuable comments, efforts, and time. We updated the draft according to the reviewers' comments and colored updated parts blue. We address each of your comments in detail as follows.
>
> **Proof sketch.** As the reviewer suggested, we added Section 5 sketching the proof of Theorem 1 including detailed discussion on motivation and intuition behind the proof in the revised draft.
>
> **Replacing $\Theta$ to big-O notation.** We thank the reviewer for pointing this out. According to the reviewer's suggestion, we replaced $\Theta(\cdot)$ with the big-O notation in the revised draft.
>
> **Implications for self-supervised learning.** We thank the reviewer for suggesting implications of our results for self-supervised learning tasks.
> We also think that our results have interesting connections with self-supervised learning.
> First, as the reviewer mentioned, self-supervised learning tasks often have much more training samples than classification tasks, e.g., predicting the image rotation (0, 90, 180, and 270 degrees) quadruples the number of training images. Hence, our results can drastically reduce the sufficient number of parameters for memorizing training samples for self-supervised learning tasks compared to existing results. In addition, a recent result showed that self-supervised learning can provably help downstream tasks under the assumption that a network can perfectly memorize self-supervised training samples (Lee et al., 2020). We think that establishing more connections between memorization and self-supervised learning would be an interesting future direction.
>
> Lee et al., "Predicting What You Already Know Helps: Provable Self-Supervised Learning" arXiv 2020

---

### Author Response · Authors · 2020-11-22
**Summary of revision**

Dear reviewers and AC,

We express our deepest gratitude for your constructive feedback and valuable comments.

In response to the questions and concerns you raised, we have carefully revised and enhanced the draft with the following additional discussions.

- Proof of Theorem 1 (Section 5)
- $\Delta$-separateness of Gaussian random vectors (Section C)

The revisions made are marked with blue in the revised draft.

Thanks,

Authors

---

### Decision · Program_Chairs · 2021-01-07
**Final Decision**

**Decision:**

Reject

**Comment:**

The AC, the reviewers, and the authors had many discussions about the results in the paper during the discussion period. Below is a brief summary.

1. The paper shows that with $O(N^{⅔})$ parameters, a feedforward neural network can memorize $N$ inputs with arbitrary labels if the inputs satisfy some mild assumptions.

2. AC brought up in the discussion phase two central questions (one of which has been raised by R3 as well)

a. The results rely on using the infinite precisions of real values in the neural networks. The results wouldn’t hold if the precision of the neural nets is finite. The subtlety about the infinite precision was not prominently discussed in the paper.

b. It’s unclear to the AC what’s the practical implication of the results to generalization or optimization. In particular, it’s unclear to the AC what a finite-sample memorization result within infinite precisions would entail. The AC thinks there is a fundamentally big difference between expressivity and finite-sample expressivity. Expressivity is a very important topic to study, whereas the motivation for studying finite sample expressivity with infinite precision is unclear. (This is raised by R3 in the reviews as well.)

3. R1 supports the paper with the following main points (The AC rephrased these with some approximations, and might misinterpret to a certain degree.)

a. The paper’s result is surprising and mathematically non-trivial.

b. Memorization is an important question to study. Many prior works study it, e.g., for showing tight VC dimension bound. It can be considered as an established setting.

c. Relying on the infinite dimension is not uncommon in ML theory.

4. R2 does not object R1’s point 3a, but seems to have a reservation to strongly recognize the technical significance of the results because it seems potentially likely to obtain the results by combining existing methods. Both R2 and the AC had some (partial) arguments to obtain the results of the paper with non-standard architecture or non-standard activations (which doesn’t subsume the paper’s results because the paper uses standard activations and feedforward net). This does make the AC unwilling to strongly recognize the technical significance of the result, but the AC doesn’t think the results are trivial either. In any case, this issue is not among the main concerns of the AC.

5. Regarding 3b, the AC thinks that unlike the prior work, the memorization results in this paper do not have an implication to the VC dimension (and in return, the dependency on $N$ is better), and this makes the significance and impact of the result in this paper somewhat unclear.

6. In summary, because the paper’s average score is somewhat borderline and because the AC has the concern 2a and 2b and was not quite convinced by the R1’s points or the authors’ responses, the AC is recommending rejection for the paper. The AC personally thinks the paper’s result has a strong potential and with additional clarification for the subtlety in 2a and additional results on the connections to generalization or optimization, the paper can be a strong one for future ML venues.